# Radiogenomic signatures reveal multiscale intratumour heterogeneity associated with biological functions and survival in breast cancer

Ming Fan [1], Pingping Xia[1], Robert Clarke [2], Yue Wang [3✉] & Lihua Li [1✉]

Advanced tumours are often heterogeneous, consisting of subclones with various genetic alterations and functional roles. The precise molecular features that characterize the contributions of multiscale intratumour heterogeneity to malignant progression, metastasis, and poor survival are largely unknown. Here, we address these challenges in breast cancer by defining the landscape of heterogeneous tumour subclones and their biological functions using radiogenomic signatures. Molecular heterogeneity is identified by a fully unsupervised deconvolution of gene expression data. Relative prevalence of two subclones associated with cell cycle and primary immunodeficiency pathways identifies patients with significantly different survival outcomes. Radiogenomic signatures of imaging scale heterogeneity are extracted and used to classify patients into groups with distinct subclone compositions. Prognostic value is confirmed by survival analysis accounting for clinical variables. These findings provide insight into how a radiogenomic analysis can identify the biological activities of specific subclones that predict prognosis in a noninvasive and clinically relevant manner.

[1] Institute of Biomedical Engineering and Instrumentation, Hangzhou Dianzi University, 310018 Hangzhou, Zhejiang, China. [2] Hormel Institute, University of Minnesota, Austin, MN 55912, USA. [3] Department of Electrical and Computer Engineering, Virginia Polytechnic Institute and State University, Arlington, VA 22203, USA. ✉email: yuewang@vt.edu; lilh@hdu.edu.cn

Cellular and molecular heterogeneity is common in solid tumours. Human cancers contain divergent genomic subclones in varying proportions, each with different gene expression patterns and biological functions[1,2]. Genomic heterogeneity contributes to adverse clinical outcomes including resistance to therapy[3] and decreased overall survival (OS)[4,5]. Characterizing the heterogeneity of tumours and complex tissues is a major challenge[6]. Deconvolution methods applied to gene expression profiles can estimate the underlying genomic subclones[7,8]. For example, data deconvolution applied to tumour/stroma gene expression profiles identified an immune subclone that is associated with decreased survival of cancer patients[9,10]. Decomposed genomic subclones are often relevant to cancer treatment, as has been shown for some immune checkpoint blockade therapies[11]. Nonetheless, significant improvements in tools and workflows are urgently needed to identify clinically and biologically meaningful genomic subclones from heterogeneous tumours that can better inform prognosis and guide treatment decisions.

Current gene expression deconvolution approaches use reference profiles in a mixture of tissue, where gene expression patterns associated with different disease states are explored in a basis matrix[12]. However, these methods cannot identify unknown biomarkers that better characterize tumour heterogeneity. Variability in the reference matrix can introduce bias into the results obtained from specific cancer types. Many known molecular signatures available to support supervised deconvolution can also be limiting and incomplete or even unreliable when the tumour microenvironment changes. In marked contrast, unsupervised approaches can extract known subclones in an unbiased manner and discover additional yet unknown subclones without the need for prior information. Transcriptional heterogeneity is modelled to identify molecularly unique genomic subclones from within the gene expression profiles of heterogeneous tumour samples[13]. Whether the molecular signatures and/or compositions of genomic subclones (including unknown subclones) identified by an unsupervised deconvolution of tumour gene expression profiles can be used either to understand cancer biology and/or to support noninvasive imaging-based prognosis is underexplored.

While most recent studies of tumour heterogeneity have focused on the molecular characterization of tissue samples, these techniques require the collection of invasive biopsies. Radiomics can be used to mine high-throughput quantitative and noninvasive image features to improve cancer diagnosis and treatment[14]. Intratumour heterogeneity can be captured by 3D imaging of the entire tumour and surrounding parenchyma[15,16]. Radiogenomics explores the relationships between radiomic features and genomic characteristics, with the goal of noninvasively uncovering relevant features that reflect the underlying biological functions most strongly associated with clinical outcomes. For example, prior studies have reported the quantitative analysis of computed tomography (CT) images in the setting of non-small cell lung cancer (NSCLC)[17,18] and magnetic resonance imaging (MRI) of tumour-adjacent parenchyma[19]. In these examples, imaging features were associated with both gene expression profiles and patient survival. A recent study identified an imaging biomarker of the genomically determined abundance of CD8 T cells that could be used to predict clinical outcomes in patients treated with immunotherapy[20]. Despite these initial efforts, the clinical value of using imaging features to determine robust prognostic genomic subclones is uncertain.

We hypothesize that the unsupervised deconvolution of gene expression profiles could reveal genomic subclones that affect cancer-related biological functions and patient survival and that radiogenomic signatures at the imaging level could capture the underlying intratumour heterogeneity evident at the molecular level. In multiscale modelling of biological systems, the term multiscale refers to the use of data from more than one scale[21]. We have used data from two scales, specifically transcriptome data and imaging data. Tumour subclone heterogeneity is identified at the genomic (transcriptome) scale using a fully unsupervised deconvolution method applied to gene expression profiles. Imaging scale intratumour heterogeneity is characterized using a set of radiomic feature analyses. The noninvasive radiogenomic signatures associating these two scales are further identified to classify patients into groups with distinct subclone compositions. Here, we investigate the biological and clinical relevance of modelling multiscale intratumour heterogeneity by conducting a radiogenomic analysis of 1310 samples of breast cancer patients on five datasets of three data cohorts (Figs. 1 and 2). The analysis is performed in three phases. First (phase 1), genomic subclones are identified using a fully unsupervised deconvolution analysis of gene expression profiles. Biological functions of the subclones are inferred by gene set enrichment analysis (GSEA); prognostic genomic signatures are identified using a genomic development dataset and tested on a genomic testing dataset. Second (phase 2), radiogenomic signatures are established by mapping radiomic features onto compositions of prognostic subclones in an independent dataset containing matched imaging and gene expression data from each tumour. Third (phase 3), the prognostic value of the identified radiogenomic signatures is further tested on another two independent datasets containing imaging and survival outcomes data. Our results provide a noninvasive and reproducible method for identifying tumour genomic subclones and their underlying biological functions that are useful for clinical applications.

## Results

**Phase 1 Tumour subclone identification.** Genome-level intratumour heterogeneity was identified by unsupervised deconvolution of gene expression data and the results were used to develop prognostic genomic signatures representing the key subclones embedded in intratumor heterogeneity. For each patient, tumour subclones were identified from tumour gene expression profiles by finding the geometric vertices and resident genes of a $J$-dimensional polytope, where $J$ is the number of subclones present in the specimen[22] (see Methods). A fully unsupervised convex analysis of mixture (CAM) method[13,23] was applied to deconvolute the gene expression profiles in the genomic development dataset ($n = 660$) from The Cancer Genome Atlas (TCGA). Grouped samples were used independently (exclusively) in different stages/steps of this study (Fig. 2). This approach generated the reference gene expression matrix that represents the overall expression value of the subclone-specific marker genes and the fraction matrix that represents the proportion of each genomic subclone over all the subclones for each sample. The number of subclones was detected using the minimum description length (MDL). MDL selected $K = 7$ as the number of subclones, representing the maximum number of sources with the minimum value (Fig. 3a).

**Biological function annotation of subclone-specific genes.** In each of the 7 genomic subclones, enriched pathways for the subclone-specific marker genes were identified by GSEA (corrected $p$ values < 0.05). A detailed description of the pathways is presented in Supplementary Table 1. Each genomic subclone was annotated by the most significantly enriched pathway uniquely present (Table 1). Cancer-related pathways included cell cycle[24], ECM-receptor interaction[25] and primary immunodeficiency[26]. The cell cycle subclone had the highest fraction value, accounting for ~22% of all subclones (Fig. 3b). Pathways enriched in cell cycle-specific marker genes included cell cycle, DNA replication,

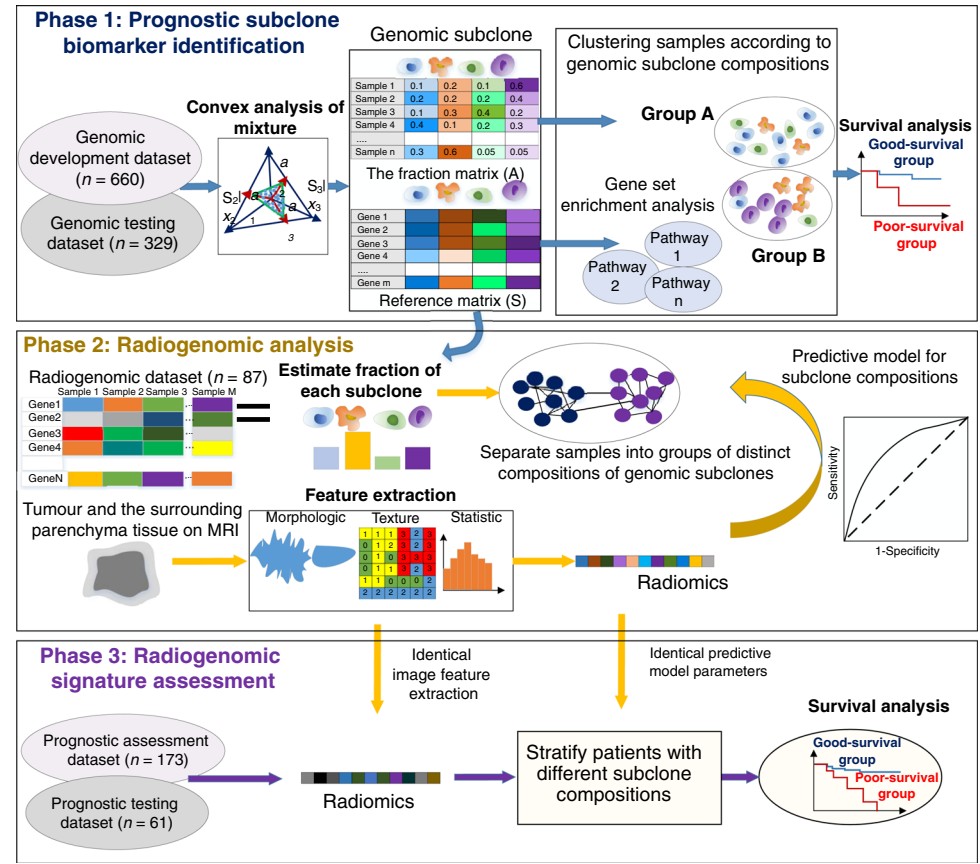

**Fig. 1 Framework of the study with three phases.** First (phase 1), prognostic subclone biomarkers are identified using a fully unsupervised deconvolution analysis of gene expression profiles. Biological functions of the subclones are inferred by gene set enrichment analysis. Second (phase 2), radiogenomic signatures are established by mapping radiomic features onto compositions of prognostic subclones. Third (phase 3), radiogenomic signatures are assessed on another two independent datasets containing imaging and survival outcomes data.

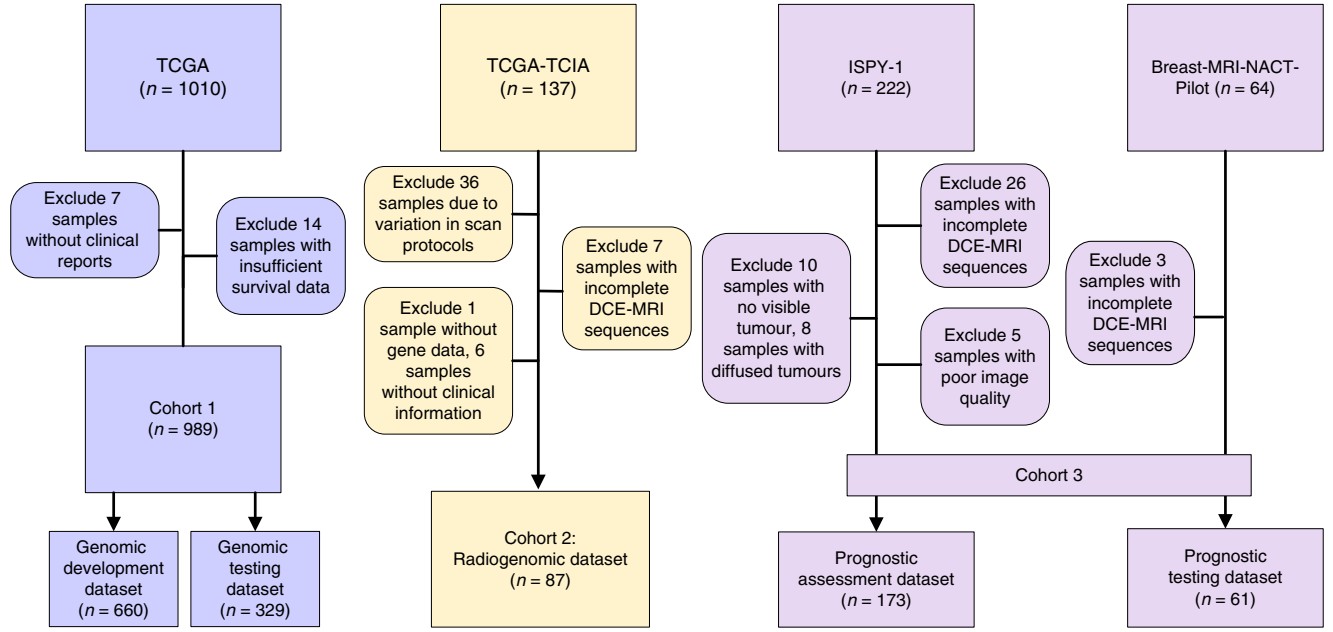

**Fig. 2 Data organization flowchart of three data cohorts.** A total of 1310 samples of breast cancer patients were included in five datasets of three data cohorts according to the selection criteria.

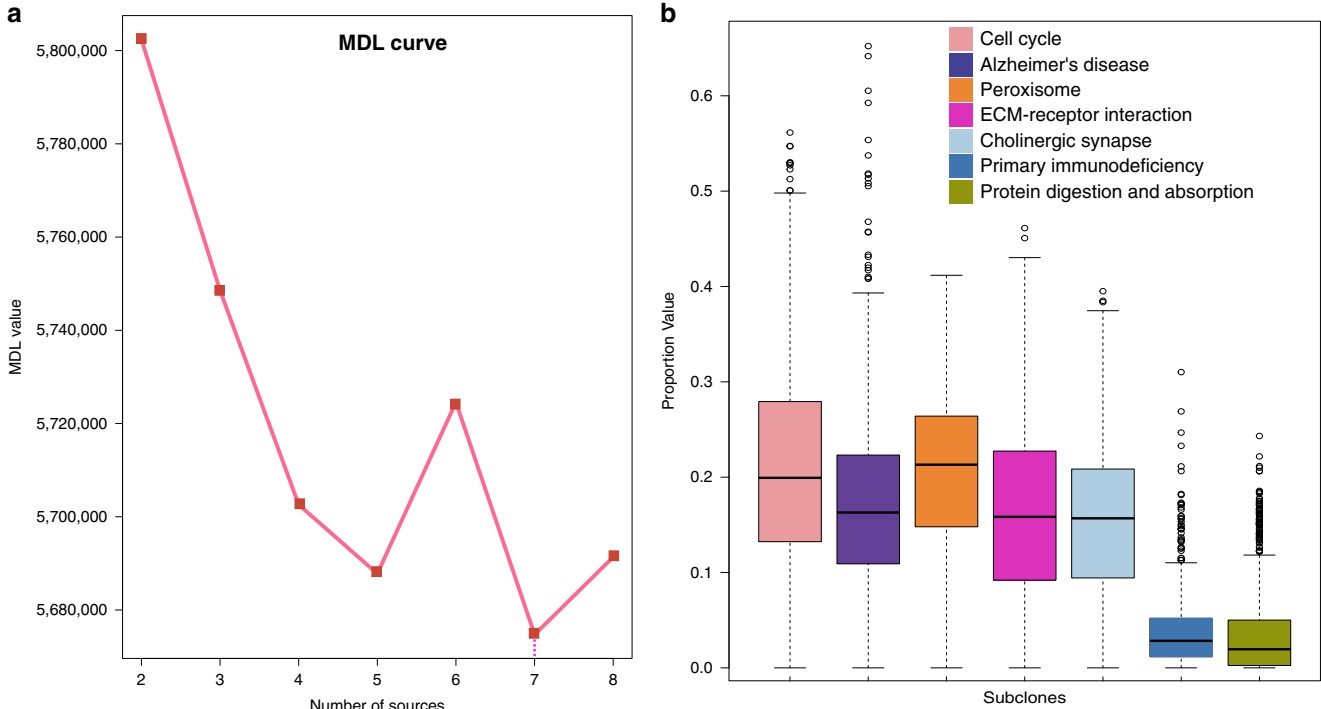

**Fig. 3 Genomic subclone decomposed by the CAM method. a** Minimum description length (MDL) curve for the gene expression decomposition based on the CAM method. **b** Proportional representation of genomic subclone fractions ($n = 660$). In the boxplot, centre line represents median; box limits indicate upper and lower quartiles; whiskers represent 1.5 × interquartile range and points are outliers. Source data are provided as a Source data file.

| Index | Number of total genes | Key function | Fraction (95% CI) | Key pathway genes | Corrected *p* value |
|---|---|---|---|---|---|
| 1 | 71 | Cell cycle | 0.220 (0.211–0.228) | *CCNB2\|CCNA2\|PLK1\|MCM2\|CDC25B\|CDK1\|MCM6\|E2F1\|PTTG1* | 1.446e-10 |
| 2 | 66 | Alzheimer's disease | 0.177 (0.170–0.185) | *BAD\|NDUFA13\|NDUFA11\|NDUFA3* | 0.018 |
| 3 | 68 | Peroxisome | 0.205 (0.199–0.212) | *PEX11A\|EPHX2\|CROT\|ACOX2* | 0.001 |
| 4 | 40 | ECM-receptor interaction | 0.163 (0.156–0.170) | *ITGA11\|ITGA5\|LAMA4* | 0.004 |
| 5 | 43 | Cholinergic synapse | 0.155 (0.148–0.161) | *ITR1\|ADCY1\|BCL2* | 0.011 |
| 6 | 29 | Primary immunodeficiency | 0.039 (0.036–0.043) | *CD3D\|IL7R\|CD3E\|PTPRC\|IL2RG* | 5.132e-09 |
| 7 | 26 | Protein digestion and absorption | 0.040 (0.036–0.044) | *COL9A3\|COL11A2\|KCNK5* | 0.0004 |

**Table 1 Genomic subclone characteristics in the genomic development dataset.**

Statistical significance was assessed by the two-sided hypergeometric test.
*p* values were adjusted for multiple comparisons.
*HR* hazard ratio, *CI* confidence interval.

progesterone-mediated oocyte maturation, oocyte meiosis, p53 signalling, mismatch repair, and pyrimidine metabolism. Cell cycle and DNA replication pathways likely reflect cell proliferation and may be used for prognostication[27]. The primary immunodeficiency subclone included signalling pathways related to immunity, such as primary immunodeficiency, T cell receptor signalling[28], natural killer cell-mediated cytotoxicity, Jak-STAT signalling[29], and immune-related B cell receptor signalling. Taken together, these results show that an unsupervised deconvolution of gene expression data can unbiasedly extract and confirm several predictive subclones that are associated with known cancer-related pathways.

**Tumour genomic-subclone assessment for prognosis.** The prognostic power of each tumour subclone was evaluated using its fraction of the total subclones to determine its correlation

(Table 2) with OS in the genomic development dataset ($n = 660$). The cell cycle and primary immunodeficiency subclones stratified patients into poor and good OS groups with significantly different survival rates (Fig. 4a, $p = 0.037$; Fig. 4b, $p = 0.0042$, respectively). Tumours containing the cell cycle subclone at >18.4% were more likely to exhibit poor OS, whereas tumours containing the primary immunodeficiency subclone with a prevalence of >3.55% were more likely to exhibit good OS.

Patient samples were further clustered by their subclone composition using consensus clustering[30]. The number of clusters that produced the most stable consensus matrix runs was selected as optimal[30]. Subsequently, patients were clustered into two groups, good-survival ($n = 316$) and poor-survival ($n = 344$), based on the cell cycle and primary immunodeficiency prognostic subclone compositions. OS rates differed between the clustered

**Table 2 Tumour subclones for survival analysis in the genomic development dataset.**

| Genomic subclone | HR (95% CI) | Log-rank p | Corrected p |
|---|---|---|---|
| Cell cycle | 1.574 (1.045–2.371) | 0.037 | 0.099 |
| Alzheimer's disease | 1.507 (0.978–2.321) | 0.085 | 0.170 |
| Peroxisome | 1.374 (0.834–2.265) | 0.257 | 0.300 |
| ECM-receptor interaction | 1.303 (0.785–2.163) | 0.268 | 0.300 |
| Cholinergic synapse | 0.751 (0.467–1.206) | 0.272 | 0.300 |
| Primary immunodeficiency | 0.519 (0.343–0.784) | 0.004 | 0.016 |
| Protein digestion and absorption | 1.249 (0.807–1.934) | 0.300 | 0.300 |
| Consensus Cluster | 1.993 (1.327–2.994) | 0.0012 | 0.010 |

Statistical significance was assessed by the two-sided log-rank test.
p values were adjusted for multiple comparisons.
HR hazard ratio.

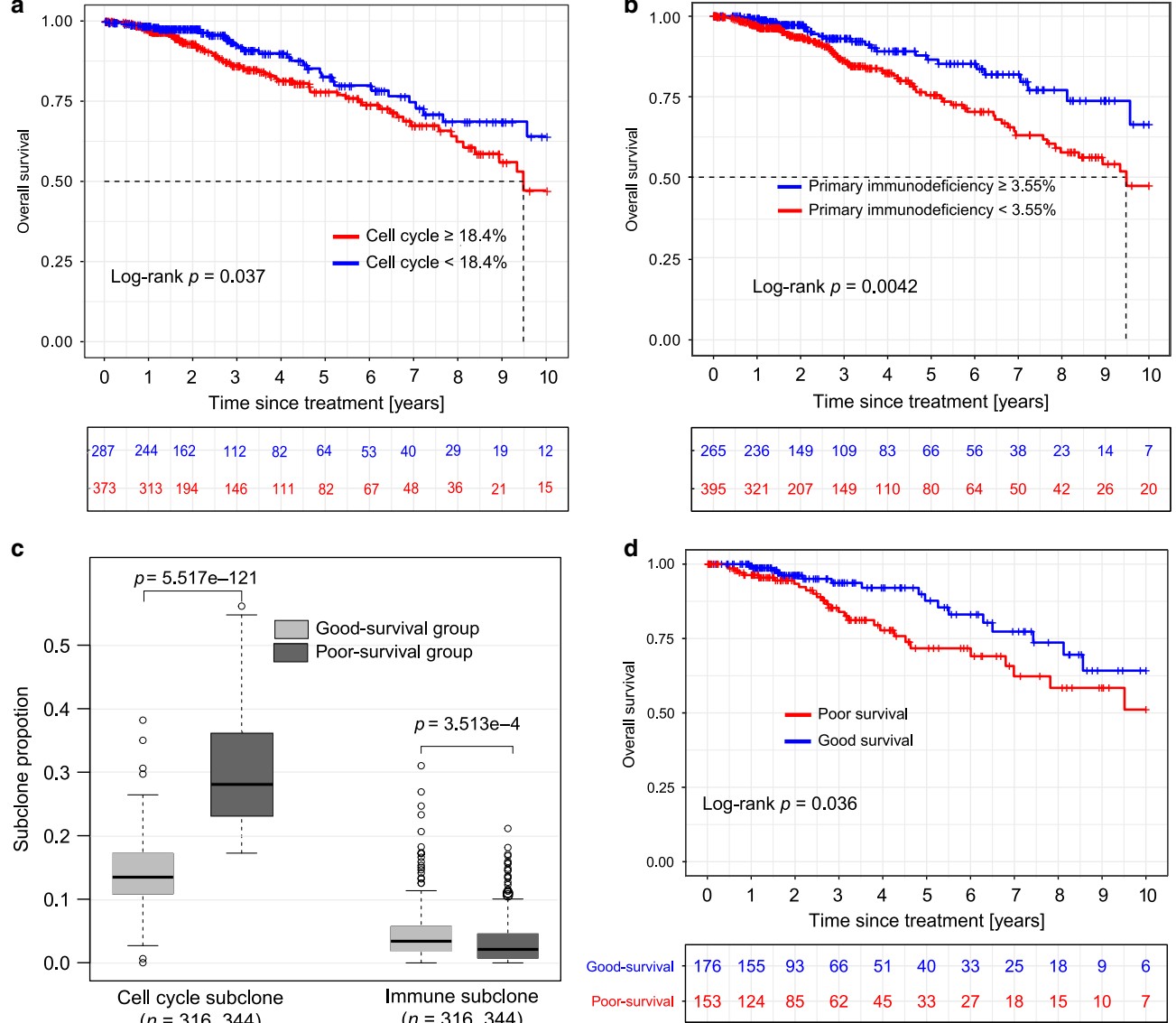

**Fig. 4 Genomic subclones correlate significantly with patient survival.** Kaplan–Meier curves of OS for the cell cycle ($p = 0.037$, two-sided log-rank test) (**a**) and primary immunodeficiency ($p = 0.0042$, two-sided log-rank test) (**b**) genomic subclones. **c** Proportional compositions of key predictive subclones of the cell cycle and primary immunodeficiency. Cell cycle subclone: $t = -29.223$, $p = 5.517\text{e-}121$, two-tailed Student's $t$ test. Immune-subclone: $t = 3.593$, $p = 3.513\text{e-}4$, two-tailed Student's $t$ test. Patients were grouped into good-survival ($n = 316$) and poor-survival ($n = 344$) according to the estimated fractions of the two subclones. **d** Kaplan–Meier curves of overall survival between good survival ($n = 153$) and poor survival ($n = 176$) groups in the genomic testing dataset ($p = 0.036$, two-sided log-rank test). Red and blue plots represent poor survival and good survival groups, respectively. In the boxplot, centre line represents median; box limits indicate upper and lower quartiles; whiskers represent 1.5 × interquartile range and points are outliers. Source data are provided as a Source data file.

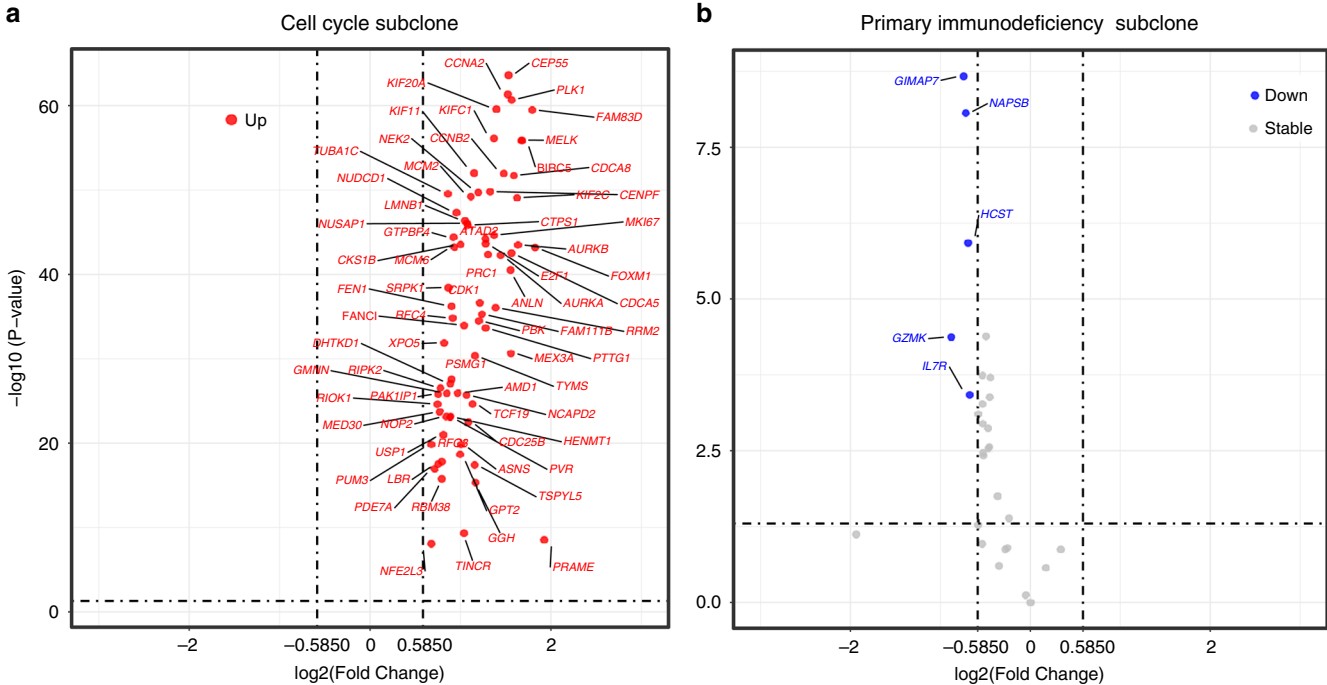

**Fig. 5 Illustration of differentially expressed genes of subclones between the poor-survival and good-survival groups.** Volcano plots are shown for the cell cycle (**a**) and primary immunodeficiency (**b**) subclones. Source data are provided as a Source data file.

patients; a smaller $p$ value ($p = 0.0012$) was obtained compared with the $p$ value for either the immunodeficiency or cell cycle subclone alone. In the poor-survival group, a significantly higher proportion of the cell cycle subclone was present when compared with the good-survival group (Fig. 4c, $p = 5.517e-121$). In contrast, tumours associated with the primary immunodeficiency subclone were present at a higher fraction in the good-survival group than in the poor-survival group (Fig. 4c, $p = 3.513e-4$).

Genes that were differentially expressed between the poor and good survival groups are shown in a volcano plot using a threshold of fold-change (FC) > 1.5 and $p < 0.05$ for the cell cycle and primary immunodeficiency subclones (Fig. 5a, b). In the poor-survival group, genes of the cell cycle subclone were upregulated, while genes in the primary immunodeficiency subclone were downregulated. Functional annotation of the differentially expressed genes of subclones between the poor- and good-survival groups for the two subclones is shown in Supplementary Fig. 1. Pathways in the cell cycle subclone were activated, since the associated genes were significantly expressed only in the poor survival group. Conversely, pathways in the primary immunodeficiency subclone were inactivated because these genes were downregulated only in the poor survival group. The expression of *CDK1* mRNA was upregulated in the cell cycle subclone. *CDK1* is associated with several cancer-related cell proliferation pathways[31] including cell cycle, DNA replication, and p53 signalling (Supplementary Figs. 1a and 2a, c). For the primary immunodeficiency subclone, downregulation of *IL7R* mRNA was associated with poor survival. *IL7R* is also associated with Jak-STAT and PI3K-Akt signalling, haematopoietic cell lineage, and cytokine-cytokine receptor interactions (Fig. 5b and Supplementary Fig. 2b, d) and is essential for normal T cell development and homeostasis[32].

**Prognostic relevance of the compositions of key subclones.** The prognostic value of the tumour subclones was evaluated using the genomic testing dataset (n = 329). The gene expression profile

was deconvoluted using the reference matrix obtained from the genomic development dataset, which generated the subclone fraction values. Patients were clustered into significantly different good survival (n = 153) and poor survival (n = 176) groups (Fig. 4d, $p = 0.036$) using the proportional compositions of key predictive subclones. Tumours with a high fraction of cell cycle and a low fraction of primary immunodeficiency subclones correlated with poor survival. Thus, intratumour genomic subclone heterogeneity is strongly associated with breast cancer survival.

**Phase 2 Radiogenomic mapping between imaging and subclones.** Radiogenomic signatures were identified as a surrogate for the prognostic genomic signatures by associating radiomic features with these genomic signatures of intratumor heterogeneity. The images and their corresponding gene expression profiles in the radiogenomic dataset (n = 87) were obtained from TCGA-BRCA of The Cancer Imaging Archive (TCIA) cohort (see "Methods"). The gene expression profile was deconvoluted under supervision, where the subclone proportion in each tumour was estimated using the reference gene expression matrix in the genomic development dataset. Patients were grouped by consensus clustering using the fractions of the cell cycle and primary immunodeficiency subclones by the same procedure applied to the genomic development dataset.

Radiogenomic signatures were extracted from dynamic contrast-enhanced MR imaging (DCE-MRI) data and used to train a predictive model for classifying patients into the poor-survival (n = 40) or good-survival (n = 47) groups (see Methods). The predictive model used the optimal feature subset (n = 15) and produced an area under the receiver operating characteristic (ROC) curve (AUC) value of 0.833, with a sensitivity of 0.723 and a specificity of 0.950. The detailed optimal feature subsets are provided in Supplementary Table 2. Among these features, nine were obtained from the tumour and six were obtained from the surrounding tumour parenchyma. Texture features such as the sum entropy and maximum probability are predictive of

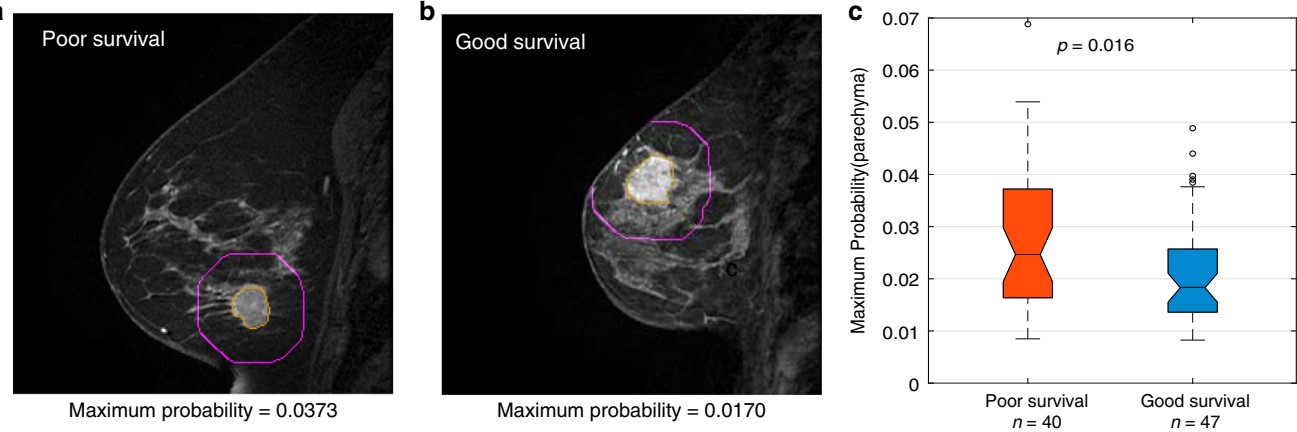

**Fig. 6 Imaging features used to predict the outcomes of patients with different breast tumour subclone compositions.** The regions of interest (ROIs) for the parenchyma region (purple line) and the tumour are shown. Examples of patients in the poor-survival group (**a**) with a feature (maximum probability) value of 0.0373 and in the good-survival group (**b**) with a feature value of 0.0170. **c** The feature values are significantly different between the good-survival ($n = 47$) and poor-survival ($n = 40$) groups (two-tailed Student's $t$ test; $t = -2.45$, df = 85, $p = 0.016$). In the boxplot, centre line represents median; box limits indicate upper and lower quartiles; whiskers represent 1.5 × interquartile range and points are outliers. Source data are provided as a Source data file.

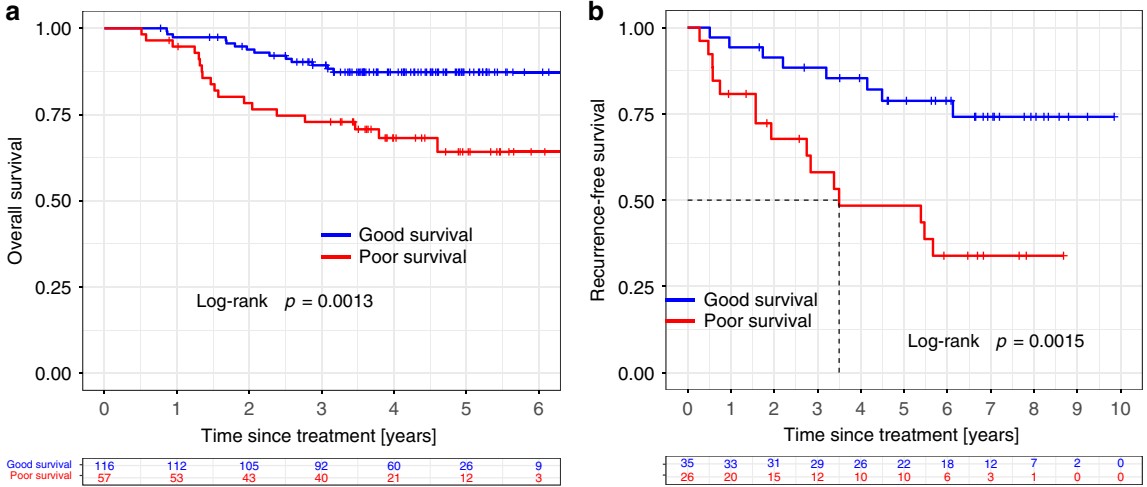

**Fig. 7 Radiogenomic signatures validated as predictors of breast cancer prognosis. a**, **b** Kaplan–Meier curves of survival in the prognostic assessment ($p = 0.0013$, two-sided log-rank test) and prognostic testing ($p = 0.0015$, two-sided log-rank test) datasets for overall survival and recurrence free survival, respectively. Red and blue plots represent poor survival and good survival groups, respectively.

subclone compositions. Figure 6 shows an example of the MR images used to illustrate a feature value (maximum probability) in the parenchyma region surrounding the tumour from S-1 (subtraction image from the precontrast and third postcontrast images). The feature value is high (Fig. 6c, $p = 0.016$) in the poor-survival group (Fig. 6a) but low in the good-survival group (Fig. 6b). Thus, radiogenomic signatures can be used as surrogates for genomic subclones.

**Phase 3 Prognostic assessment of radiogenomic signatures.** The clinical significance of the radiogenomic signatures was assessed by determining their ability to predict survival. Two independent datasets (prognostic assessment and prognostic testing) with matched DCE-MRI and survival data were obtained from the NACT-pilot and ISPY-1 cohorts, respectively, in TCIA. The trained predictive model and the radiogenomic signatures from the second phase were used to classify patients into the poor or good survival groups. Differences in either OS or recurrence-free survival (RFS) between the two groups were compared to evaluate the prognostic significance of the radiogenomic signatures.

Radiogenomic signatures clearly discriminate between patients with poor ($n = 57$) or good ($n = 116$) OS (Fig. 7a) in the prognostic assessment dataset (log-rank $p = 0.0013$). These signatures were further tested for RFS in the prognostic testing dataset by separating patients into the poor- ($n = 26$) and good-survival ($n = 35$) groups (Fig. 7b; $p = 0.0015$). Each of these studies independently validated the effectiveness of the radiogenomic signatures as predictors of breast cancer prognosis.

To examine whether the proposed radiogenomic biomarkers are independent of the available clinical variables including age, race, oestrogen receptor (ER) status, progesterone receptor (PR) status, human epidermal growth factor receptor (HER2) status, we conducted multivariate survival analyses on both the prognostic assessment and prognostic testing datasets (Table 3).

First, after adjusting the model for the clinical variables, the proposed radiogenomic signatures maintained independent significance for both OS ($p = 0.0067$; prognostic assessment dataset) and RFS ($p = 0.0037$; prognostic testing dataset). Second, including the proposed radiogenomic signatures in the model as an explanatory variable, the experimental results on the prognostic assessment dataset showed a significantly increased

**Table 3 Multivariate analysis of radiogenomic signatures and clinical variables.**

| Clinical variables and radiogenomic signature | Prognostic assessment dataset | | Prognostic testing dataset | |
|---|---|---|---|---|
| | HR (95% CI) | Wald *p* value | HR (95% CI) | Wald *p* value |
| Age | 0.99 (0.94–1) | 0.57 | 0.95 (0.89–1) | 0.057 |
| ER | 0.17 (0.051–0.54) | 0.0028 | 0.31 (0.07–1.4) | 0.12 |
| PR | 1.5 (0.45–4.7) | 0.52 | 0.97 (0.21–4.5) | 0.97 |
| HER2 | 1.3 (0.6–2.8) | 0.50 | 0.31 (0.086–1.1) | 0.077 |
| Tumour size | 1 (1–1) | 0.07 | 1.3 (1–1.7) | 0.037 |
| Radiogenomic signature | 2.8 (1.3–5.8) | 0.0067 | 7.7 (1.9–31) | 0.0037 |

Statistical significance was assessed by the two-sided Wald test.
*ER* oestrogen receptor, *PR* progesterone receptor, *HER2* human epidermal growth factor receptor, *HR* hazard ratio.

($p < 1e-6$) model fit for OS ($p = 0.00014$) compared with using only the clinical variables ($p = 0.0010$). The consistent increase in prognostic power was replicated in the prognostic testing dataset. The multivariate survival model showed a significantly increased ($p = 0.0012$) correlation with RFS after inclusion of the radiogenomic signatures ($p = 0.0029$) compared with using only the clinical variables ($p = 0.02$). These results showed that our prediction model using the proposed radiogenomic signatures has additional and independent prognostic power after accounting for these clinical variables.

## Discussion

The intratumoural heterogeneity of genomic subclones in breast tumours was identified using an unsupervised gene expression deconvolution method. Cancer-related biological functions of the genes that defined the cell cycle and primary immunodeficiency subclones provided mechanistic insights into the subclone associations with OS outcomes in breast cancer patients. Radiogenomic signatures were extracted to establish a link between imaging heterogeneity and genomic signatures of intratumour heterogeneity. The prognostic implication of the imaging signatures was validated for their association with survival in independent datasets. Our proposed prognostic radiogenomic signatures are clinically feasible and complementary to the clinically relevant information available for most patients and therefore have the potential to improve the accuracy of clinical prognostication.

Solid tumours are composite ecological systems with interacting components that can create a unique pathophysiological state. It may not be able to be fully represented in biopsy samples that are subject to sampling bias and can miss key properties of the tumour microenvironment[33]. This ground state is dynamic, affected by key features of the multiscale cell survival decision network present in tumour cells, and deeply reflected in the molecular landscape of intratumour heterogeneity[34]. While a few studies have reported that radiomic and/or genomic signatures may be predictive of survival status[35,36], their translation into routine clinical practice has been limited mainly due to the following two reasons. On the one hand, medical imaging provides a noninvasive and tumour-wide method to view the entire tumour ecosystem. However, conventional radiomic signatures associated only with clinical follow-up are information-poor and lack the key refinement provided by knowledge of the critical molecular characteristics reflective of intratumour heterogeneity. On the other hand, the deep phenotyping based on the predictive genomic signatures, which are derived collectively, averaged across a large number of highly heterogeneous tumour samples and are then correlated only with clinical follow-up data, are information-rich[37]. However, the data are obtained using a local, invasive method that cannot adequately capture essential information about the entire tumour ecosystem and are therefore imprecise for individual patients and often clinically impractical where repeated biopsies are needed.

To address the critical problem of the absence of signatures enriched with prognostic information deeply embedded in intratumour heterogeneity, our predictive model is built using radiogenomic signatures that capture the hidden associations between the prognostic genomic signatures of intratumour heterogeneity and radiomic features. Here, genomic signatures represent the mixing patterns of the key predictive subclones that constitute tumour heterogeneity. The rich information deeply rooted in these key subclones, which is predictive of tumour survival outcomes, is uncovered by an unsupervised deconvolution of intratumour heterogeneity using genomics data. The resulting genomic signatures, which reflect the proportional compositions of key predictive subclones, are then used to develop the radiogenomic signatures by associating these genomic signatures with radiomic features. Lastly, the predictive radiogenomic signatures and survival data are used to build a clinically useful prediction model.

While multigene assays (such as Mammaprint[38], Oncotype DX[39], Prosigna[40]) are increasingly used in clinical breast cancer prognostication, their overall prognostic accuracy is somewhat limited. These assays rely on biomarker expression patterns derived from a population of tumour biopsies and applied to a single biopsy (or very small number of biopsies) from each individual patient. Since core biopsy sampling rarely adequately captures information about the entire tumour ecosystem, these invasive methods can be imprecise and are often difficult to repeat in individual patients. In contrast, we have developed predictive and noninvasive radiogenomic signatures by enriching the radiomic features (derived from clinical imaging) with a priori predictive genomic signatures (derived from key predictive subclone compositions). Moreover, the imaging-based tests can be performed repeatedly over time, allowing for longitudinal studies that are often challenging when multiple repeated biopsies would otherwise be required of patients.

Previously, a study in patients with NSCLC associated imaging features with prognostic gene expression patterns[18]. Our study differs notably from this prior work because genomic subclones were analysed to explore heterogeneity within the tumour. Additionally, a recent study evaluated the abundance of CD8 cells by applying a supervised expression profile method to genomic data. CT radiomic signatures were extracted to determine their association with immune-related molecular biomarkers and used to evaluate responses to immunotherapy[20]. In marked contrast, we applied a fully unsupervised decomposition method to bulk tumour gene expression profiles that requires no prior information regarding the number or marker genes of the subclones. Furthermore, our unsupervised gene expression profile deconvolution method also identified an immune-related subclone associated with a favourable prognosis. While another study

identified gene expression signatures for predicting prognostic imaging features, the effectiveness of the identified gene signatures for prognosis was evaluated by leveraging public gene expression profiles and survival data[19]. In contrast, we incorporated biological and clinical insights to derive the prognostic biomarker of the subclone from the tumour gene expression profiles. The three phases of the unique radiogenomic analysis strategy provide a noninvasive imaging surrogate for probing molecular-level tumour heterogeneity.

The gene expression deconvolution method builds upon fundamental mathematical theorems[13] that were initially applied to study tissue heterogeneity in human coronary and aortic atherosclerosis[41]. Notably, a reference gene expression matrix that represents the overall gene expression level of the subclones was estimated with an unsupervised CAM method (first phase). This reference matrix was further used as a guide for supervised gene expression profile deconvolution to validate genomic subclones that accurately predict prognosis. The genomic subclone was reanalysed using the reference matrix associated with aberrant biological functions and poor survival (second phase). This approach decreases bias in the gene expression deconvolution method that may arise from variability in cancer types or gene expression platforms. To design more stable and flexible models targeting patient outcomes, additional gene expression profile data that represent comprehensive cancer-related functions should be included.

Tumours with a high proportion of the cell cycle subclone and a low proportion of the primary immunodeficiency subclone were associated with poor survival. Notably, subclones with the most significantly enriched pathways also had the strongest correlation with survival. The cell cycle subclone had the highest proportion among all the genomic subclones and was correlated with poor survival. Among the signalling pathways within this subclone, the cell cycle and DNA replication pathways likely reflect cell proliferation and may be used for prognostication[27]. A related radiogenomic study revealed an association between tumour radiomics and gene expression patterns and showed enriched pathways of the cell cycle in head and neck cancer patients[15]. Overexpression of cell cycle genes and prevalence of the cell cycle subclone further support a central role for increased cell growth as a driver of poor survival.

Among the pathways definitive of the primary immunodeficiency subclone profile, the primary immunodeficiency pathway was most enriched. A previous study reported that genes upregulated in this pathway were associated with prolonged relapse-free survival[42], which is broadly reflective of the results reported here. Others reported that the inactivated Kyoto Encyclopedia of Genes and Genomes (KEGG) pathways we detected, namely, cytokine-cytokine receptor interaction, haematopoietic cell lineage, and Jak-STAT signalling pathway, were also altered in lung cancer subtypes[43]. Genes defining the primary immunodeficiency pathway were significantly enriched in hepatocellular carcinomas (HCCs) and correlated with the density of Epstein-Barr virus (EBV)-positive tumour-infiltrating lymphocytes (TILs)[44], suggesting that the immunodeficiency pathway may reflect the degree of immune cell infiltration in tumours. Breast tumours lacking immune infiltration are associated with a poor prognosis[5], consistent with our data showing that tumours with a high fraction of immune-related subclones were associated with improved OS.

The Alzheimer's diseases subclone includes pathways with mitochondrial genes as annotated by the pathway analysis tools. Down-regulation of the NDUFA13 gene in the pathways has been shown to correlate with lymph node metastasis and advanced tumour-node-metastasis (TNM) stage in breast cancer[45]. The peroxisome subclone includes peroxisome, prostate cancer, pathways in cancer, metabolic pathways. Among these, the peroxisome pathway is reported as enriched among all breast tumour subtypes[46] and is correlated with cell proliferation and tumorigenesis[47,48]. The extracellular matrix (ECM)-receptor interaction subclone includes ECM-receptor pathway, which is associated with progression and resistance to cytotoxic and hormonal treatments in breast cancer[49]. Moreover, the dysregulated PI3K-AKT signalling pathways in this subclone are often activated by mutations in breast cancers and so can be targeted by drugs[50]. The ECM-receptor interaction and the protein digestion and absorption pathways are reported as significantly enriched in analysis of differentially expressed genes between tumour tissues and paracancerous tissues[51]. The cholinergic synapse subclone includes key gene of BCL2, which is an established prognostic biomarker in early breast cancer[52].

Unlike prior studies, we derived prognostic radiogenomic signatures from tumour gene expression rather than requiring that the analysis be supervised by a directional association with patient outcomes[53]. Thus, imaging technology can aid in our understanding of biological functions in tumours that drive breast cancer patient outcomes. For example, we show that a high feature value of maximum probability for the parenchymal region surrounding the tumour, which reflects high intratumour heterogeneity, correlates with a high cell cycle fraction and a low immunodeficiency fraction. Radiogenomic analyses, which are repeatable and noninvasive, may offer a powerful surrogate for difficult-to-repeat invasive genomic analyses.

The quality of the MRI data used in this study may be suboptimal because they were collected in the past, their use was an unavoidable necessity because of the requirement for up to a decade of clinical follow-up to collect the OS outcomes data. The follow-up time for data from TCGA (Cohorts 1 and 2) is shorter than that of Cohort 3 but we do not see a significant impact on the main outcomes of this work. In our study, Cohorts 1 and 2 are used solely to identify the prognostic radiogenomic features by associating the multiscale key components of intratumour heterogeneity, that contributes mechanistically to the poor outcomes in patients. To achieve this unique objective, we selected TCGA/TCIA dataset for the following two reasons. First, TCGA/TCIA is currently the largest publicly available dataset containing the matched genomic and the clinical follow-up information needed to complete the study. Second, this dataset provides the matched genomic and DCE-MRI data that allow us to uncover clinically useful associations between radiomic features and genomic signatures. The innovative data analytics methodology and pipeline developed here are readily applicable to more recently obtained, higher-quality MRI data for further validation once adequate clinical follow-up information becomes available.

In conclusion, the current study identified the intratumoural heterogeneity of genomic subclones with cancer-related biological functions and prognostic power. This molecular-level heterogeneity was modelled noninvasively by radiogenomic signatures that reflect imaging heterogeneity. The radiogenomic strategy we developed using an unsupervised method to identify genomic subclones can be extended to other cancers to explore tumour heterogeneity and for further use in clinical practice.

## Methods

**Datasets.** The study was approved by the institutional review board (IRB) at Hangzhou Dianzi University (IRB-2019001). In the three-phase analysis, 1310 samples were used for (1) genomic subclone identification and prognostic power assessment; (2) radiogenomic analysis of associations between DCE-MRI features and genomic subclones; and (3) radiogenomic signature assessment and validation for prognosis prediction (Figs. 1 and 2). The demographic and clinical metadata within the study datasets include age, race, ER status, PR status, HER2 status, histologic type, surgery type, and follow-up status (alive, dead, or lost to follow-up) and are shown in Supplementary Table 3.

Three data cohorts with unique and complementary strengths were purposely organized and used in the three-phase analysis respectively for our radiogenomic signature discovery and prediction model building pipeline. Cohort 1 contains matched genomic and follow-up data from TCGA-BRCA[54] and was used for the unsupervised gene expression deconvolution and to develop the prognostic genomic signatures that are deeply rooted in the key subclones that contribute to intratumour heterogeneity. The dataset initially included 1097 samples. Samples with matched imaging and genomic data were included in Cohort 2 ($n = 87$, after proper quality control procedure) and the remaining data ($n = 1010$) were included in Cohort 1. Seven samples were excluded because there was no clinical report, and 14 samples were excluded because they had insufficient survival data, resulting in 989 samples for analysis. We randomly separated the dataset into a genomic development dataset ($n = 660$, 66.7%) and a genomic testing dataset ($n = 329$, 33.3%). Within the genomic development dataset ($n = 660$), 567 censored patients had a mean follow-up time of 3.147 years; the remaining patients ($n = 93$) had a mean follow-up time of 3.769 years. For the genomic testing dataset ($n = 329$), 283 patients were censored with a mean follow-up time of 3.150 years and the remaining patients ($n = 46$) had a mean follow-up time of 3.801 years. Cohort 2 (radiogenomic dataset, $n = 87$) contains matched imaging and genomic data and was used to establish the associations between radiomic features and the prognostic genomic signatures (survival data are not needed). The genomic data was obtained from TCGA-BRCA and the matched imaging data was obtained from TCIA[55]. The dataset initially comprised 137 samples. Thirty-six samples were excluded due to the variation in scan protocols, one sample was excluded because of absent gene expression data, 6 samples were excluded due to missing clinical information, and 7 samples were excluded because they had incomplete DCE-MRI data. Thus, the final dataset comprised 87 patients for analysis. Cohort 3 contains matched DCE-MRI and follow-up data and includes two datasets; the prognostic assessment dataset (collected from the ISPY-1 trials in TCIA)[56] and the prognostic testing dataset (collected from the Breast-NACT-pilot in TCIA)[57,58]. The prognostic assessment dataset initially included 222 samples[56]. A total of 173 samples remained for the analyses after excluding 10 samples with no visible tumour, 8 samples with diffuse tumours, 5 samples with poor imaging quality, and 28 samples with incomplete DCE-MRI data. The prognostic testing dataset initially included 64 samples, but three samples were excluded because of incomplete imaging, leaving 61 samples for study. The optimized prediction model trained in Cohort 2 was used to classify the patient samples into good-survival and poor-survival groups in the independent test sets (prognostic assessment dataset, $n = 173$; prognostic testing dataset, $n = 61$).

**Imaging protocol.** For the radiogenomic dataset, DCE-MRI protocol was described elsewhere[59]. In brief, the imaging was performed using a T1-weighted gradient echo sequence, including one precontrast image and three to five postcontrast images after injection of the gadolinium-based contrast agent. The in-plane resolution varied from 0.53 to 0.85 mm, the slice thickness ranged from 2 to 3 mm, and the flip angle was 10°.

For the prognostic assessment dataset, DCE-MRI protocol was described elsewhere[56]. DCE-MRI was acquired sagittally at 1.5-T using a T1-weighted fat-suppressed sequence and a dedicated breast coil. A gradient echo sequence was acquired with a precontrast sequence, followed by early-phase and delayed-phase sequences at 2.5 mins and 7.5 mins after injection of the contrast material. The imaging parameters were as follows: repetition time (TR), ≤ 20 ms; echo time (TE), 4.5 ms; flip angle, ≤45°; field of view, 160–180 mm; matrix, 256 × 192; 64 slices; slice thickness, ≤2.5 mm; and in-plane resolution, ≤1 mm.

For the prognostic testing dataset, imaging protocol was described in TCIA (Breast-MRI-NACT-Pilot)(https://wiki.cancerimagingarchive.net/display/Public/Breast-MRI-NACT-Pilot). The DCE-MRI data were acquired with the patients placed in the prone position using 1.5-T MRI (GE Healthcare, Milwaukee, Wis). T1-weighted fat-suppressed MRI was performed under the following parameters: TR, 8 ms; TE, 4.2 ms; matrix, 256 × 192 × 60; flip angle, 20°; field of view, 180–220 mm; in-plane resolution, 0.7–0.9 mm; and slice thickness, 2–2.4 mm. A series of postcontrast images were obtained following the intravenous injection of a bolus of 0.1 mmol/kg gadobutrol. The time interval between the first postcontrast series and the contrast agent injection was 2.5 mins, while the time interval between the second and first postcontrast image series was 7.5 mins.

**CAM method for the decomposition of gene expression profiles.** The CAM method assumes that the mixed gene expression obtained from bulk heterogeneous tissues represents the proportional contributions of several distinct latent variables that reflect the presence of multiple genomic subclones. Hence, mixed expression profiles were decomposed based on the expression values of a group of marker genes that were exclusively enriched in each genomic subclone[60].

The CAM method can identify molecular markers directly from the original mixed expression matrix of $X$ and further estimate the subclone-specific expression profile matrix of $S$ and the constituent proportion matrix of $A$. This linear mixing model can be formulated as $X = A \times S$.

The gene expression profile for each sample $i$ is denoted as x ($i$) and can be expressed as a nonnegative linear combination of the subtype-specific component $s_k$, weighted by the relative subclone proportions $a_j (i)$ in that sample, as defined by

the following equation:

$$x(i) = \left\{ \sum_{j=1}^{J} a_j(i) s_j \Big| a_j(i) \geq 0, \sum_{j=1}^{J} a_j(i) = 1, i = 1, \cdots, N \right\} \quad (1)$$

where $a_j$ is the vector notation of the expression values of matrix $A$ and $J$ is the number of subclones.

This method first clusters genes into an optimal number of representative and robust clusters $[x_m]$, using affinity propagation clustering and expectation-maximization mixture model fitting[61]. To identify the vertices of a convex set $\mathcal{H}\{x\}$, the CAM method was used to identify the gene clusters spatially located at the corners of the clustered pixel time series scatter simplex by applying a minimum error margin convex hull for data fitting:

$$\delta_{m,[1,\cdots J] \epsilon C_j^M} = \min \left\| x_m - \sum_{j=1}^{J} a_j x_j \right\|_2, a_j \geq 0, \sum_{j=1}^{J} a_j = 1. \quad (2)$$

The most probable $J$ vertices were estimated by optimizing the sum of the margin between the convex hull and the remaining exterior cluster centres to reach the minimum:

$$\left[ g_{k=1,\ldots,K_*} \right] = \underset{[1,\cdots J] \epsilon C_j^M}{\operatorname{argmin}} \sum_{m=1}^{M} \delta_{m,[1 \cdots J] \epsilon C_j^M}. \quad (3)$$

Each sample of gene expression data can be decomposed by weighted linear combinations of the latent pattern of gene expression for a specific genomic subclone and the corresponding proportions. The indices of the subclone are based on the genes assigned to the gene cluster at a vertex. A model is selected with the optimal $K$ subclone, while $K$ is determined by the MDL as defined by Chen et al.[62].

On the basis of the expression levels of subclone-specific marker genes detected by the CAM method, the relative proportions of constituent subtypes are estimated using standardized averaging. Based on the reference matrix $A$ and the marker genes, the proportion matrix $S$ is estimated from matrix $S$.

**Image processing and feature analysis.** DCE-MRI data were normalized and resized[63]. Specifically, pixel values were normalized by the mean of the interquartile values of the gland tissue in the precontrast MR image, ensuring that the signals inside the tumour and from the parenchymal region were normalized comparably. DCE-MRI data were resized to the same resolution prior to feature extraction with a common in-plane resolution of 0.8 mm and a common slice thickness of 2 mm. The centre of the tumour was first annotated by a radiologist with more than 10 years of experience. The breast tumour was segmented through a spatial fuzzy C-means (FCM) algorithm to segment a volumetric breast tumour from the background tissue[64]. The DCE-MRI series was registered to reduce the effect of possible body movement during imaging[65]. Fibroglandular tissue that surrounded the tumour was segmented using FCM clustering, excluding the skin and fatty tissue from the breast tissue. Imaging tissue that was 2 cm wide surrounding the tumour was identified. Specifically, peritumoural shell that surrounds the tumour margin were identified by manual annotation[16].

Feature extraction was performed for each sample; details are provided in Supplementary Table 4. Radiomic features were extracted from the tumour and the surrounding tissue because they can provide information on both the tumour and the microenvironment[66–68]. We extracted features inside these regions on the image series of the precontrast, the subtraction images between the second postcontrast image series and precontrast series, and between the late postcontrast and precontrast series, which were termed S-0, S-1 and S-2, respectively. We obtained 10 histogram-based features, 10 morphologic-based features, and 19 Haralick features that measure textural heterogeneity based on the grey-level co-occurrence matrix (GLCM). Statistical features were calculated using standard statistics based on the histogram of pixel values. Texture features were extracted using the publicly available, open-source and validated PET Oncology Radiomics Test Suite (PORTS) package[69]. Both the standard histogram-based and Haralick features were obtained for S-0, S-1, and S-2. Morphological features were evaluated on S-0 using standard shape analysis methods. Comparative experiments using our method and Pyradiomics software produced almost identical sets of statistical and morphologic features. All image processing and feature extraction processes were performed in MATLAB (R2015, MathWorks, Natick, Mass).

**Data analysis.** Our study framework was performed in three phases on four datasets (Figs. 1 and 2). In the first phase, genomic subclones were identified by applying the CAM method to the gene expression profiles. The biological function and prognostic value of each subclone were assessed. Based on the estimated proportion of each subclone, patients were clustered into groups representing good and poor survival, in which the differences in survival rates were compared and validated. In the second phase, genomic subclones were identified by gene expression deconvolution analysis in an additional dataset with available DCE-MRI data and corresponding gene expression profiles. Thereafter, the radiogenomic signatures were identified from the tumour and parenchyma images to determine their association with the subclone compositions, thus predicting survival. In the third phase, we assessed the prognostic value of the radiogenomic signatures on two independent datasets with DCE-MRI and corresponding survival data.

For each genomic subclone, biological mechanisms were identified with pathway analyses of the marker genes that were exclusively enriched in the subclone[60]. The hypergeometric test was used to evaluate the significance of the marker genes enriched in each cell subclone with reference to KEGG pathways. GSEA was used to identify enriched biological pathways using the KEGG pathway database. The volcano plot and the display of the relationship between genes and terms for the illustrations of differentially expressed genes were obtained using R GOplot.

Kaplan–Meier analysis was used to visualize differences in RFS and OS. A log-rank test assessed significant differences between curves; the cutoff point was determined by the optimal threshold value that yielded the smallest log-rank $p$ value. The hazard ratio (HR) and its 95% confidence interval were obtained to assess the differences in survival between the stratified groups. A multivariate Cox regression model was used to determine whether the radiogenomic features were independently associated with OS or RFS after adjusting for the available clinical variables of age, ER, PR, HER2 and tumour size. Patients who did not have an event by the end of 10 years were censored at that time. A likelihood ratio test was used to evaluate whether a survival model including the radiogenomic signatures as an explanatory variable significantly improves the model fit compared with a survival model using only the clinical variables.

To build an optimal model with limited samples, we conducted cross validation-based optimization in each of the major steps in our model building pipeline. In addition to the general rule of thumb for controlling model complexity, we also considered the complex interactions (joint-effect) among the radiomic features and the orthogonal, complementary nature of the texture, morphological, and statistical features. We conducted extensive 10-fold cross validation procedures to optimize both feature selection and model building (reducing potential underfitting or overfitting). To address the collinearity problem in subsequent variable selection, correlation-based clustering of the initial features was used to remove redundant features and to select the most representative radiomic features. A global optimization step used the Evolutionary Algorithm to select the most informative radiomic features to build our prediction models[64]. A support vector machine with a radial basis function kernel was used as the base classifier in this global optimization step. L1 regularization was applied to the base classifier to control for complexity. Prediction performance was evaluated using the AUC (the area under the ROC curve) criterion and 10-fold cross validation. The cutoff ROC value was determined at the maximum Youden index (sensitivity + specificity − 1).

To control for false discovery rates (FDRs), the Benjamini-Hochberg method was used in the survival analysis for individual subclones. FDR-corrected $p$ values < 0.1 were considered significant. All statistical analyses were performed using MATLAB (R2015, MathWorks, Natick, MA, USA) and R (version 4.0 R Foundation for Statistical Computing, Vienna, Austria).

**Reporting Summary**. Further information on research design is available in the Nature Research Reporting Summary linked to this article.

## Data availability

The gene expression data of Cohort 1 (Genomic development and Genomic testing datasets) and Cohort 2 (Radiogenomic dataset) are available from TCGA-BRCA project of Genomic Data Commons [https://portal.gdc.cancer.gov/projects/TCGA-BRCA]. The matched imaging data of Cohorts 1 and 2 are available from TCGA-BRCA in TCIA on website [https://wiki.cancerimagingarchive.net/display/Public/TCGA-BRCA]. The imaging, clinical and survival data of the Prognostic testing dataset are available from the Breast-MRI-NACT-Pilot in TCIA on website [https://wiki.cancerimagingarchive.net/display/Public/Breast-MRI-NACT-Pilot]. The imaging, clinical and survival data of the Prognostic assessment dataset are available from the ISPY-1 trail in TCIA on website [https://wiki.cancerimagingarchive.net/display/Public/ISPY1]. Source data are provided with this paper. The remaining data are available within the article, Supplementary Information or available from the author upon request.

## Code availability

Codes for genomic subclone identification and validation were available on [Github].

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

## Acknowledgements

This work was supported by the National Natural Science Foundation of China (61731008, 61871428), the Natural Science Foundation of Zhejiang Province of China (LJ19H180001), and the US National Institutes of Health (CA184902 and HL111362-05A1).

## Author contributions

M.F. and L.L. designed the study, analysed the data and wrote the paper. P.X. contributed to the study design, image processing, gene expression data analysis and statistical analysis. R.C. participated in the bioinformatics methods and was a major contributor in reviewing and editing the paper. Y.W. participated in the initial design and contributed to designing the gene expression decomposition method. All authors read and approved the final paper.

## Competing interests

The authors declare no competing interests.
