## [Peer Review File · Nature Communications]

Reviewers' comments:

Reviewer #1 (Remarks to the Author):

1. IRB/informed consent: can they authors please provide this information?

2. Add more detailed information on survival data.

3. Radiomics methodology- more detail is necessary:

*Please explain how many grey levels the data was reduced to (for improved counting statistics) and whether any form of data normalization was performed prior to feature calculations, with the purpose of removing very low/high pixel intensities.

Please provide information on what software you have used. If it is in-house the authors need to be clear about that and also whether their software has been validated against more widely available packages such as MaZda and/or Pyradiomics.

*There seems to be some confusion in nomenclature here by the authors. A training set (=TCGA) is use to develop a model, a validation set for fine-tuning, and finally a test set to assess performance. The test set must never have been seen previously, hence the alternate name of holdout set. Having these three distinctions is often seen in neural network development. As far as I understand they've used logistic regression so I don't see why you need a validation dataset. I think they've got a training set (TCGA) and two test sets.

*I would also question their methods in terms of calculating features (especially histogram based features) from raw data that hasn't been normalized. So, as far as I can tell they've calculated subtraction images (S1 and S2) without dividing by the precontrast image. This makes the comparison of features such as mean, range, minimum, maximum etc across patients highly dubious.

*Both the in-plane resolution and slice thickness vary for their training cohort (and also in their assessment and validation cohorts). Have they considered interpolating the data to all have the same resolution, prior to feature calculations?

*The model has too many parameters. The authors got 24 parameters with 47 cases of poor survival and 40 cases of good survival in the training set, so their model is overparametrized. There is no hard rule but I'd say around 8-10 parameters is the maximum they should consider. Ten parameters would be 4 cases per parameter in the good survival group (people quote a minimum of 3-10 cases for each parameter per class).

*They also need to provide more information as to how the logistic regression (this is a pretty basic technique and surely it would be better to employ non-linear methods such as support vector machines?) is performed. I think they've got $(10+10+19) \times 3 = 117$ parameters here. There are, from a practical perspective, almost an infinite number of ways you can extract n parameters (where n goes from 1 to 117) from 117 to build models with, so how did they arrive at the models/parameters they've used? This is not a straightforward problem!

4. I think it has to be clearly stated that the quality of the MRI data used is no longer state of the art/out-dated. I am aware that the studies used, i.e TCG, I-SPY provide access to large numbers of genomic data but since the conduction of the MRI experiments there have been significant improvements in MRI software and hardware. It has to be addressed/discussed that using such suboptimal imaging e.g 1st post-contrast time point 2:30 min after injection, the results of this study may not be representative of the potential of MRI radiomics in this context.

Reviewer #2 (Remarks to the Author):

This paper looks at radiomic analysis of MRI images to predict heterogeneity and overall survival in breast cancer patients. It is a solid paper with a good design and uses multiple publicly available datasets. The idea, while not new, has never before been performed in breast cancer and the results are very promising.

The paper is very well written and logical. I recommend publication with no major corrections.

My only hesitation are the small #'s in the TCIA dataset and I couldn't find the tumor types that were analyzed.

Reviewer #3 (Remarks to the Author):

The authors should elaborate how their proposed method would be used clinically?

More discussion is needed about the prognosis/follow-up data available in the TCGA cohort. It is suspected that follow-up is limited in this cohort and that there are very few survival events, therefore it is not clear how this cohort can be representative of breast cancer transcriptomic survival signatures.

Many studies have been published on the use of molecular biomarkers that predict overall survival in breast cancer. In particular there are several gene expression signatures for breast cancer (MammaPrint, OncotypeDX) that are clinically being used. More importantly several clinical features have been described that are very important for predicting prognosis of breast cancer including but not limited to age, stage, ER, PR, HER2, lymph node status etc. The authors have to study how their proposed deconvolution biomarkers add additional prognostic correlation on top of these existing prognostic biomarkers.

The TCGA-TCIA-BRCA is a subset of the genomic TCGA-BRCA data set, where the patients with imaging data (N=87) removed from the TCGA-BRCA data set (Genomic development and validation data set)?

None of the analysis on prognosis are correcting the models for common clinical features such as age, size, grade, stage, hormone receptor status etc. It is important to know if the radiogenomic biomarkers are independent of these clinical features.

Specific comments:

Figure 3, the y-axis is not labeled.

Figure 4d is a training set performance, and if so should not be reported.

Figures 5 & 6 provide too extensive details about the enrichment analysis of the two subclones, cell cycle & immune. These figures can be condensed & portions moved to supplementary.

Figure 7d, is this performance based on a training set, it is not clear what the evaluation strategy was for this model? If so it cannot be reported as it is not a representation of unseen data.

Responses to the review comments and summary of the revisions

Manuscript ID: NCOMMS-20-02749

Title: Radiogenomic signatures reveal multiscale intratumour heterogeneity associated with biological functions and survival in breast cancer

By: Ming Fan, Pingping Xia, Robert Clarke, Yue Wang, and Lihua Li

We greatly appreciate and are highly encouraged by the supportive comments and constructive suggestions provided by each of the three expert reviewers. We wish to thank the reviewers and the Associate Editor for giving us the opportunity to revise, improve, and resubmit our manuscript for further consideration.

In the revised manuscript, we have addressed each of the reviewer's concerns, clarified any areas of confusion, added new experimental results and discussions, and improved both the figures and the overall presentation of our study. The suggestions from the reviewers have helped us to improve both our work and our manuscript substantially.

Below, we provide detailed responses to each of the reviewers' comments, identifying where and how each comment has been addressed in the revised manuscript. Since some reviewers raised similar concerns, we have addressed these in detail in response to the reviewer who raised the issue first and cite back to this response when it is raised again by another reviewer. Changes in the revised manuscript are marked by **red-colored** text.

In summary, we have made the following primary revisions:

1. Additional experimental results and discussion are now provided to show that the proposed radiogenomic signatures enriched with genomic subclone biomarkers indeed add additional prognostic values to that provided by the available clinical variables (age, ER status, PR status, HER2 status, and tumour size).
2. Additional experimental results and discussion are now provided to show that, using the confounder-adjusted survival models, the proposed radiogenomic signatures maintain their prognostic power independent of the available clinical variables.
3. Image normalization and data interpolation have been properly performed. All relevant text and experimental results (*e.g.*, radiomic feature extraction and predictive model building) have been updated accordingly. Importantly, the updated experimental results do not change our main findings and conclusions.
4. New text with more details are added to both the study description and discussion, clarifying the main objective of this work and justifying how TCGA/TCIA datasets were used.

5. More information is provided to describe how the prediction models were developed and optimized including the multistep radiomic feature selection noted above.

6. Additional experimental results and discussion are provided to elaborate how our proposed method would be used clinically.

7. More details and discussion are provided to explain how the three data cohorts were organized and used in this study including both the data types and prognostic/follow-up information.

Below we provide detailed responses to each of the reviewer's comments, identifying where and how each comment has been addressed in the revised manuscript. As noted above, where reviewers raised similar concerns, these are addressed in response to the reviewer who raised the issue first. We cite back to this response when the issue is raised again. Changes in the revised manuscript are marked by **red-colored** text.

Answers to the comments by Reviewer #1

We thank the reviewer for the supportive comments on our work.

1. *IRB/informed consent: can they authors please provide this information?*

We have now added the information requested on the relevant institutional review board (IRB) approval by Hangzhou Dianzi University (IRB-2019001).

2. *Add more detailed information on survival data.*

The available survival data (alive, dead, or lost to follow-up) are provided in Supplementary Table S3.

3. *Please provide information on what software you have used. If it is in-house the authors need to be clear about that and also whether their software has been validated against more widely available packages such as MaZda and/or Pyradiomics.*

The gray-level co-occurrence matrix (GLCM) based textures were extracted using publicly available radiomic software, *i.e.*, the open-source and validated PET Oncology Radiomics Test Suite (PORTS) package [1]. Comparative experiments using PORTS and Pyradiomic produce almost identical sets of GLCM-based texture features.

Statistical features (*e.g.*, the mean, median, skewness, and kurtosis) were calculated using standard statistics based on the histogram of pixel values. Morphological features were calculated using standard shape analysis methods. Comparative experiments using our method and Pyradiomics produce almost identical sets of statistical and morphologic features.

We now cite the reference describing PORTS [1] and clarify the software tools used to calculate the radiomic features from the DCE-MRI data. These descriptions have been added in the "Image processing and feature analysis" section.

[1] Larry P. PET Oncology Radiomics Test Suite 3D Image Texture Metric Calculation Package (<https://www.mathworks.com/matlabcentral/fileexchange/55587-ports-3d-image-texture-metric-calculation-package>), Version 1.1; 2016

4. *There seems to be some confusion in nomenclature here by the authors. A training set (=TCGA) is use to develop a model, a validation set for fine-tuning, and finally a test set to assess performance. The test set must never have been seen previously, hence the alternate name of holdout set. Having these three distinctions is often seen in neural network development. As far as I understand they've used logistic regression so I don't see why you need a validation dataset. I think they've got a training set (TCGA) and two test sets.*

We regret that our previous description was unclear. In this study, three data cohorts with unique and complementary strengths were specifically organized and selectively used in the different stages of our radiogenomic signature discovery and prediction model building pipeline.

Cohort 1 contains matched genomic and follow-up data from TCGA and was randomly separated into a genomic development dataset (n=660) and a genomic testing dataset (n=329) (this 'genomic testing' dataset was the 'genomic validation' dataset in the previous submission). Cohort 1 was used for unsupervised gene expression deconvolution and to develop the prognostic genomic signatures that are deeply rooted in the key subclones that contribute to intratumour heterogeneity. The prognostic genomic signatures were trained using the genomic development dataset and then tested on the genomic testing dataset.

Cohort 2 contains matched imaging and genomic data (radiogenomic dataset, n=87), where the genomic data were obtained from TCGA-BRCA, and the matched imaging data were obtained from The Cancer Imaging Archive (TCIA). This cohort was used to establish the associations between radiomic features and the afore-established prognostic genomic signatures. The radiogenomic signatures were identified as a surrogate for the prognostic genomic signatures. Please note that we used a 10-fold cross validation procedure to optimize both feature selection and model building (*i.e.*, reducing potential model underfitting or overfitting).

Cohort 3 contains matched DCE-MRI and follow-up data and includes two datasets; the prognostic assessment dataset (collected from the ISPY-1 trials in TCIA) and the prognostic testing dataset (collected from the Breast-NACT-pilot in TCIA). The optimized prediction model trained in Cohort 2 was used to classify the patient samples into good-survival and poor-survival groups in the independent test sets, *i.e.*, the prognostic assessment dataset (n=173) and prognostic testing dataset (n=61) (this 'prognostic testing' dataset is the 'prognostic validation' dataset in the previous submission).

In the revised manuscript, we have used clearer terms for the steps in the data analytics pipeline as discussed above. These descriptions have been added in the "Datasets" section.

5. I would also question their methods in terms of calculating features (especially histogram based features) from raw data that hasn't been normalized. So, as far as I can tell they've calculated subtraction images (S1 and S2) without dividing by the precontrast image. This makes the comparison of features such as mean, range, minimum, maximum etc across patients highly dubious.

We agree with the reviewer. In the revised manuscript, DCE-MRI data have been normalized as suggested and the relevant experimental results have been updated accordingly. Specifically, pixel values were normalized by the mean of the interquartile values of the gland tissue in the precontrast MR image, ensuring that the signals inside the tumour and from the parenchymal region were normalized comparably. Please note that image normalization was performed using the same procedure as in a previous study [2]. In the revised manuscript, we added a description of this image normalization procedure to the "Image processing and feature analysis" section.

[2]. Wu J, Li X, Teng X, et al. Magnetic resonance imaging and molecular features associated with tumour-infiltrating lymphocytes in breast cancer. *Breast Cancer Research*: 2018; 20:101.

6. Both the in-plane resolution and slice thickness vary for their training cohort (and also in their assessment and validation cohorts). Have they considered interpolating the data so all have the same resolution, prior to feature calculations?

We have performed appropriate data interpolations as suggested. All imaging data now have the same resolution prior to feature extraction. Specifically, the common in-plane resolution is 0.8 mm, and the common slice thickness is 2 mm. All relevant text and experimental results (e.g., radiomic feature extraction; predictive model building) have been updated accordingly. Importantly, the updated experimental results do not change our main findings and conclusions.

In the revised manuscript, we have added descriptions of the image interpolation procedure in the "Image processing and feature analysis" section.

7. The model has too many parameters. The authors got 24 parameters with 47 cases of poor survival and 40 cases of good survival in the training set, so their model is overparametrized. There is no hard rule but I'd say around 8-10 parameters is the maximum they should consider. Ten parameters would be 4 cases per parameter in the good survival group (people quote a minimum of 3-10 cases for each parameter per class).

We share the same concern as the reviewer regarding the potential for model overfitting when working with relatively small sample sizes. Precisely for this reason, i.e., to build an optimal model with limited samples, we conducted cross validation-based optimization in each of the major steps in our model building pipeline. In addition to the general rule of thumb for controlling model complexity, we also considered the complex interactions (joint-effect) among the radiomic features and the orthogonal, complementary nature of the texture, morphological, and statistical features. We conducted extensive 10-fold cross validation procedures to optimize both the feature selection and model building (reducing potential

underfitting or overfitting). This has produced an optimized prediction model that now contains only 15 parameters as determined by the most informative feature subspace.

We have added new text to the discussion to clarify and/or justify the relevant steps in our model building pipeline in the “Data analysis” section.

8. *They also need to provide more information as to how the logistic regression (this is a pretty basic technique and surely it would be better to employ non-linear methods such as support vector machines?) is performed. I think they've got (10+10+19)x3 = 117 parameters here. There are, from a practical perspective, almost an infinite number of ways you can extract n parameters (where n goes from 1 to 117) from 117 to build models with, so how did they arrive at the models/parameters they've used? This is not a straightforward problem!*

As suggested by the reviewer, we have included results using a support vector machine with a nonlinear kernel as the prediction model. In building these models, we were fully aware of the potential model overfitting problem and therefore conducted parameter regularization and cross validation experiments to optimize both the features and models.

We designed a multistep optimization pipeline to obtain the optimal feature subset for building the prediction models. The initial feature extraction step generated 194 radiomic features, collectively derived from the imaging data covering both tumour and the surrounding parenchyma regions. To address the collinearity problem in subsequent variable selection step, correlation-based clustering of the initial features was conducted to remove redundant features and to select the most representative radiomic features (n=100). Lastly, we used the Evolutionary Algorithm for global optimization and identified the final 15 most predictive radiomic features to build our prediction models. A support vector machine with a radial basis function kernel was used as the base classifier in this global optimization step. L1 regularization was applied to the base classifier to control for complexity. Prediction performance was evaluated using the AUC (area under the ROC curve) criterion and 10-fold cross validation.

In the revised manuscript, we have provided more details on how the prediction models were developed and optimized, including the multistep feature selection noted above in the “Data analysis” section.

9. *I think it has to be clearly stated that the quality of the MRI data used is no longer state of the art/out-dated. I am aware that the studies used, i.e TCG, I-SPY provide access to large numbers of genomic data but since the conduction of the MRI experiments there have been significant improvements in MRI software and hardware. It has to be addressed/discussed that using such suboptimal imaging e.g 1st post-contrast time point 2:30 min after injection, the results of this study may not be representative of the potential of MRI radiomics in this context.*

While we agree with the reviewer that the quality of the MRI data used in this study may be suboptimal because it was collected in the past, their use was an unavoidable necessity because of the requirement for up to a decade of clinical follow-up to collect the overall survival outcomes data. However, the main objective of this work was to identify prognostic and global radiomic features that reflect the molecular landscape of intratumour heterogeneity that contributes to poor outcomes in patients. To achieve this unique objective, we selected TCGA/TCIA dataset for the following two reasons. First, TCGA/TCIA is currently the largest publicly available dataset containing the matched genomic and clinical follow-up information needed to complete the study. Second, this dataset provides the matched genomic and DCE-MRI data that allow us to uncover novel and clinically useful associations between radiomic features and genomic signatures. The innovative data analytics methodology and pipeline developed here are readily applicable to more recently obtained, higher-quality MRI data for further validation once adequate clinical follow-up information becomes available.

In the revised manuscript, we have added new text with more details to both the study description (in the “Dataset” section) and the discussion to clarify the main objective of this work and to justify our use of TCGA/TCIA dataset (in the “Discussion” section).

Answers to the comments from Reviewer #2

We thank the reviewer for the supportive comments on our work.

1. This paper looks at radiomic analysis of MRI images to predict heterogeneity and overall survival in breast cancer patients. It is a solid paper with a good design and uses multiple publicly available datasets. The idea, while not new, has never before been performed in breast cancer and the results are very promising. The paper is very well written and logical. I recommend publication with no major corrections.

We thank the reviewer for these supportive comments.

2. My only hesitation are the small #'s in the TCIA dataset and I couldn't find the tumor types that were analyzed.

Please note that we provided the tumour histologic types (e.g., invasive ductal IDC, infiltrating lobular carcinoma, and others) in Supplementary Table S3 in our original submission. In the revised manuscript, we have added new text to clarify the tumour types in this study. The information was added as follows: “The demographic and clinical data in the study datasets, including age, race, ER status, PR status, HER2 status, histologic type, surgery type, and follow-up status (e.g., alive, dead, or lost to follow-up), are given in Supplementary Table S3.”

While we shared the same concern as the reviewer regarding the relatively small number of informative cases in the TCIA dataset, we opted to use TCGA/TCIA dataset for the two reasons detailed in our response to Reviewer #1. Please also note that the predictive radiomic

features derived from TCGA/TCIA dataset were validated further on two independent cohorts with adequate samples (n=61 and n=173 samples, respectively) and sufficient follow-up information.

In the revised manuscript, we have provided more description of the dataset used in this study in the “Dataset” section.

Answers to the comments from Reviewer #3

We thank the reviewer for the insightful and constructive comments on our work.

1. The authors should elaborate how their proposed method would be used clinically?

This important comment deserves further elaboration. Our newly updated experimental results show that our prediction model using the proposed radiogenomic signatures has additional and independent prognostic power after adjusting for the available clinical variables of age, ER status, PR status, HER2 status, and tumour size; more details on the experimental results are provided in our response to Comment #3 (below). This additional prognostic power is achieved by associating the tumour-wide global radiomic features (noninvasive imaging) with the predictive genomic signatures of intratumour heterogeneity. Our proposed prognostic radiogenomic signatures are clinically feasible and complementary to the clinically relevant information available for most patients and therefore have the potential to improve the accuracy of clinical prognostication.

Solid tumours are composite ecological systems with interacting components that can create a unique pathophysiological state. It may not be able to be fully represented in biopsy samples that are subject to sampling bias and can miss key properties of the tumour microenvironment [3]. This ground state is dynamic, affected by key features of the multiscale cell survival decision network present in tumour cells, and deeply reflected in the molecular landscape of intratumour heterogeneity [4].

While a few studies have reported that radiomic and/or genomic signatures may be predictive of survival status, their translation into routine clinical practice has been limited mainly due to the following two reasons. On the one hand, medical imaging provides a noninvasive and tumour-wide method to view the entire tumour ecosystem. However, conventional radiomic signatures associated only with clinical follow-up are information-poor and lack the key refinement provided by knowledge of the critical molecular characteristics reflective of intratumour heterogeneity. On the other hand, the deep phenotyping based on predictive genomic signatures, which are derived collectively, averaged across a large number of highly heterogeneous tumour samples and are then correlated only with clinical follow-up data, are information-rich [5]. However, the data are obtained using a local, invasive method that cannot adequately capture essential information about the entire tumour ecosystem and are therefore imprecise for individual patients and often clinically impractical where repeated biopsies are needed.

To address the critical problem of the absence of signatures enriched with prognostic information deeply embedded in intratumour heterogeneity, our predictive model is built

using novel radiogenomic signatures that capture the hidden associations between the prognostic genomic signatures of intratumour heterogeneity and radiomic features. Here, genomic signatures represent the mixing patterns of the key predictive subclones that constitute tumour heterogeneity. The rich information deeply rooted in these key subclones, which is predictive of tumour survival outcomes, is uncovered by an unsupervised deconvolution of intratumour heterogeneity using genomics data. The resulting genomic signatures, which reflect the proportional compositions of key predictive subclones, are then used to develop the radiogenomic signatures by associating these genomic signatures with radiomic features. Lastly, the predictive radiogenomic signatures and survival data are used to build a clinically useful prediction model.

In the revised manuscript, we have provided additional experimental results in the “Phase 3: Radiogenomic signature assessment for association with prognosis” section. Moreover, we have provided more discussion to elaborate both the roadmap of this research and how our proposed method would be used clinically in “Discussion” section.

[3]. Koren S, Bentires-Alj M: Breast Tumor Heterogeneity: Source of Fitness, Hurdle for Therapy. *Mol Cell* 2015, 60(4):537-546.

[4]. K. A. McDonald, T. Kawaguchi, Q. Qi, X. Peng, M. Asaoka, J. Young, M. Opyrchal, L. Yan, S. Patnaik, E. Otsuji, and K. Takabe, Tumor Heterogeneity Correlates with Less Immune Response and Worse Survival in Breast Cancer Patients, *Ann Surg Oncol*, 26(7), 2191-2199, 2019.

[5] Cathryn M. Delude, Deep phenotyping: The details of disease, *Nature*, 527, S14–S15(2015).

2. More discussion is needed about the prognosis/follow-up data available in the TCGA cohort. It is suspected that follow-up is limited in this cohort and that there are very few survival events, therefore it is not clear how this cohort can be representative of breast cancer transcriptomic survival signatures.

We thank the reviewer for this important comment, which deserves further discussion. In this study, the three data cohorts with unique and complementary strengths were purposely organized and used in the different stages of our proposed prediction model building pipeline, resulting in the novel radiogenomic signatures. The three cohorts are described above in our response to Reviewer 1 (Comment 4). Additional clarification is provided below.

Cohort 1 was used to conduct unsupervised deconvolution and to develop the prognostic genomic signatures that represent the key subclones embedded in intratumour heterogeneity. This cohort contains the matched genomic and follow-up data from TCGA and was randomly separated into a genomic development dataset (n=660) and a genomic testing dataset (n=329) with adequate survival events (n=101 and 50, respectively). The genomic signatures were trained in the genomic development dataset and the prognostic value was tested in the genomic testing dataset.

Cohort 2 was used to establish the association between radiomic features and the afore-established prognostic genomic signatures where survival data are not needed. Our design of cross-utilizing Cohorts 1 and 2 increases the likelihood that the radiogenomic signatures are

more generalizable. The proposed radiogenomic signatures are further validated using Cohort 3.

Cohort 3 contains the matched imaging and follow-up data from TCIA and with sufficient survival events (49 recurrence and 32 death out of 173 subjects, 23 recurrence out of 61 subjects). The optimized prediction model trained in Cohort 2 was validated in Cohort 3.

The three data cohorts were used separately for training, assessment/validation, and testing. Using multiple independent cohorts for model training and testing ensures the development of more generalizable radiogenomic signatures, avoiding possible overfitting of the model. Indeed, our experimental results, supported by the validated prognostic power, further justify our design strategy for developing generalizable radiogenomic signatures.

In the revised manuscript, we have provided more detailed descriptions and discussion of the three data cohorts and how they were organized and used in this study in the “Datasets” section.

3. Many studies have been published on the use of molecular biomarkers that predict overall survival in breast cancer. In particular there are several gene expression signatures for breast cancer (MammaPrint, OncotypeDX) that are clinically being used. More importantly several clinical features have been described that are very important for predicting prognosis of breast cancer including but not limited to age, stage, ER, PR, HER2, lymph node status etc. The authors have to study how their proposed deconvolution biomarkers add additional prognostic correlation on top of these existing prognostic biomarkers.

While multigene assays (*e.g.*, MammaPrint [6], Oncotype DX [7], Prosigna [8], BCI [9], and EndoPredict [10]) are increasingly used in clinical breast cancer prognostication, their overall prognostic accuracy is somewhat limited. These assays rely on biomarker expression patterns derived from a population of tumour biopsies and applied to a single biopsy (or very small number of biopsies) from each individual patient. Since core biopsy sampling rarely adequately captures information about the entire tumour ecosystem, these invasive methods can be imprecise and are often difficult to repeat in individual patients. In contrast, we have developed predictive and noninvasive radiogenomic signatures by enriching the radiomic features (derived from clinical imaging) with *a priori* predictive genomic signatures (derived from key predictive subclone compositions). Moreover, the imaging-based test can be performed repeatedly over time, allowing for longitudinal studies that are often challenging when multiple repeated biopsies would otherwise be required of patients. In the revised manuscript, we have added these discussions in the Discussion section.

Concerning the clinical variables available in Cohort 3 (ER, ER, HER2, tumour size, and age), our newly updated results show that our radiogenomic model has additional, independent prognostic power after accounting for these clinical variables.

To demonstrate this point, we conducted multivariate survival analyses on both the prognostic assessment and prognostic testing datasets (Table 3). First, after adjusting the model for the clinical variables, the proposed radiogenomic signatures maintained independent significance for both OS ($p=0.0067$; prognostic assessment dataset) and RFS

($p=0.037$; prognostic testing dataset). Second, including the proposed radiogenomic signatures in the model as an explanatory variable, the experimental results on the prognostic assessment dataset showed a significantly increased ($p<1e-6$) model fit for OS ($p=0.00014$) compared with using only the clinical variables ($p=0.0010$). The consistent increase in prognostic power was replicated in the prognostic testing dataset. The multivariate survival model showed a significantly increased ($p=0.0012$) correlation with RFS after inclusion of the radiogenomic signatures ($p=0.0029$) compared with using only the clinical variables ($p=0.02$). These results showed that our prediction model using the proposed radiogenomic signatures has additional and independent prognostic power after accounting for these clinical variables.

In the revised manuscript, we have provided additional experimental results and discussion to further elaborate how the proposed tumour-wide radiogenomic signatures enriched with deconvolution biomarkers add additional prognostic value to that provided by available clinical prognostic variables. The additional results have been added in the “Phase 3: Radiogenomic signature assessment for association with prognosis” section in the **Results**.

[6] van 't Veer, L. J. et al. Gene expression profiling predicts clinical outcome of breast cancer. *Nature* 2002, 415, 530-536.

[7]. Kwa M, Makris A, Esteva FJ: Clinical utility of gene-expression signatures in early stage breast cancer. *Nature Reviews Clinical Oncology* 2017, 14(10):595-610.

[8]. Sestak I, Dowsett M, Zabaglo L, Lopez-Knowles E, Ferree S, Cowens JW, Cuzick J: Factors predicting late recurrence for estrogen receptor-positive breast cancer. *Journal of the National Cancer Institute* 2013, 105(19):1504-1511.

[9]. Sgroi DC, Sestak I, Cuzick J, Zhang Y, Schnabel CA, Schroeder B, Erlander MG, Dunbier A, Sidhu K, Lopez-Knowles E et al: Prediction of late distant recurrence in patients with oestrogen-receptor-positive breast cancer: a prospective comparison of the breast-cancer index (BCI) assay, 21-gene recurrence score, and IHC4 in the TransATAC study population. *The Lancet Oncology* 2013, 14(11):1067-1076.

[10]. Filipits M, Rudas M, Jakesz R, Dubsy P, Fitzal F, Singer CF, Dietze O, Greil R, Jelen A, Sevelde P et al: A new molecular predictor of distant recurrence in ER-positive, HER2-negative breast cancer adds independent information to conventional clinical risk factors. *Clinical Cancer Research* 2011, 17(18):6012-6020.

4. The TCGA-TCIA-BRCA is a subset of the genomic TCGA-BRCA data set, where the patients with imaging data (N=87) removed from the TCGA-BRCA data set (Genomic development and validation data set)?

As discussed in our response to Comment 2 above, three data cohorts with unique and complementary strengths were specifically organized and separately used in the different stages of our proposed prediction model building pipeline, resulting in the novel radiogenomic signatures. The three cohorts are described in detail above. As noted by the reviewer, TCGA-TCIA-BRCA is a subset of the genomic TCGA-BRCA dataset ($n=1097$). Samples with matched imaging and genomic data were used in Cohort 2 and were then removed from TCGA-BRCA dataset. The remaining data ($n=1010$) were used in Cohort 1. After excluding 21 samples according to our selection criteria (a detailed description of the

selection criteria is shown in the Datasets section), 989 samples were used and randomly separated into the genomic assessment (n=660) and genomic testing datasets (n=329) to identify genomic signatures and to test their prognostic power.

In the revised manuscript, we have provided more description and discussion about the three data cohorts and how they were organized and used in this study in the “Dataset” section.

5. *None of the analysis on prognosis are correcting the models for common clinical features such as age, size, grade, stage, hormone receptor status etc. It is important to know if the radiogenomic biomarkers are independent of these clinical features.*

We fully agree with the reviewer that it is important to know if the proposed radiogenomic signatures are independent of the available clinical features (*e.g.*, age, tumour size, grade, stage, hormone receptor status, *etc.*). Please see our response to Comment 3 (above). Please also note that data for grade, stage, and lymph node status are not available in Cohort 3 (the prognostic assessment and prognostic testing datasets) and could not be included in the model.

In the revised manuscript, we have provided additional experimental results and more discussion using the confounder-adjusted survival models to show that the proposed radiogenomic biomarkers are independent of these clinical features with respect to prognostic power.

Specific comments:

6. *Figure 3, the y-axis is not labeled.*

We have labelled the y-axis in Figure 3 (the minimum description length (MDL) value) in the revised manuscript.

7. *Figure 4d is a training set performance, and if so should not be reported.*

As suggested, we have removed Figure 4d.

8. *Figures 5 & 6 provide too extensive details about the enrichment analysis of the two subclones, cell cycle & immune. These figures can be condensed & portions moved to supplementary.*

As recommended by the reviewer, we have removed Figure 6 from the main text; it is now shown in Supplementary Figure S2. Figure 5 (panels c, d) was moved to Supplementary Figure S1.

9. *Figure 7d, is this performance based on a training set, it is not clear what the evaluation strategy was for this model? If so it cannot be reported as it is not a representation of unseen data.*

Figure 7d reports the ability of the radiogenomic signatures to capture the information present in the predictive genomic signatures, where tumour-wide radiomic features with the

predictive are associated with genomic signatures. The performance criterion is the area under the ROC curve obtained via cross validation on the training dataset.

In the revised manuscript, this panel has been removed.

REVIEWER COMMENTS

Reviewer #3 (Remarks to the Author):

This sentence in the introduction is not correct:

It is not clear what the meaning is of the term “multi-scale” in the title.

I disagree with this sentence:

“Whether the molecular features of genomic subclones identified by unsupervised deconvolution of tumour gene expression can be used either to understand cancer biology and/or to predict prognosis accurately is largely unknown.”

This has been extensively studied by several reports, in particular for deconvolution methods, these reports show clearly the correlation of deconvolved gene expression profiles with cancer biology, immune components and prognosis, also in breast cancer. The novelty in this study is really the use of a non-invasive biomarker that reflects the deconvolved components.

The cohort discussion is confusing in several instances in the manuscript:

- Figure 2 shows 4 cohorts, and the methods section mentions 3 cohorts. Can the authors make it consistent throughout the manuscript how many cohorts there are?
- The introduction mentions “two datasets containing both gene expression and survival data”, please specify exactly which two data sets. As far as I can tell there is only 1 such data set.
- The TCIA-TCGA breast cancer samples are a subset of the TCGA cohort, can the authors explicitly mention in the results that the TCIA-TCGA cohort samples were removed from the TCGA cohort? These samples cannot be used twice, in Phase 1 and phase2?

I do not agree with the novelty in this statement: “Taken together, these results show that an unsupervised gene expression deconvolution method can discover novel subclones associated with cancer-related pathways.” Most of these pathways have been reported at length for breast cancer, as the literature on gene signatures that are prognostic for breast cancer is very extensive and has already shown that the immune component is an important aspect for breast cancer survival prediction together with the obvious cell cycle pathways.

Further, do the authors have any explanation, rationale, interpretation for the Alzheimer’s disease component, the peroxisome component, the ECM- component and the cholinergic synapse component? These are all components that have higher importance than the primary immunodeficiency component, yet are not discussed? Especially the Alzheimer’s component is puzzling. Any thoughts what the significance is of these components in breast cancer?

I disagree that the TCGA data (this includes the TCGA cases that are used in cohort 2 that also have imaging data) has adequate survival events. For breast cancer it has been shown in several reports that 5 to 10 year follow-up data is required to be able to capture robust survival signatures across patients. The authors should mention the absolute number of censored patients and the average follow-up time for both censored and non-censored patients so the reader can judge for themselves.

Based on the data that you report, 84% of patients in both the development and testing cohorts are censored, what are the follow-up times for these?

Figure 4 d: this panel is not clear, which one is it? One of the clones?

In the last paragraph of Phase 3, the p-value for RFS ($p=0.037$) does not match the pvalue mentioned in Table 3

Responses to the reviewers' comments and summary of the revisions

Manuscript ID: NCOMMS-20-02749A

Title: Radiogenomic signatures reveal multiscale intratumour heterogeneity associated with biological functions and survival in breast cancer

By: Ming Fan, Pingping Xia, Robert Clarke, Yue Wang, and Lihua Li

We greatly appreciate the specific comments and suggestions provided by the reviewer #3. We wish to thank the reviewer and the Associate Editor for giving us the opportunity to revise, improve, and resubmit our manuscript for further consideration.

We have updated the affiliation of Dr. Clarke upon his relocation to a new institute during the preparation and revision.

In the further revised manuscript, we have addressed the reviewer's concerns, clarified any areas of confusion, added new discussions and revised figures. Below, we provide detailed responses to the reviewer's comments, identifying where and how each comment has been addressed in the revised manuscript. Changes in the revised manuscript are marked by **red-colored** text.

Reviewer #3 (Remarks to the Author):

1. ***This sentence in the introduction is not correct: It is not clear what the meaning is of the term "multi-scale" in the title.***

In multiscale modelling of biological systems, the term multiscale refers to the use of data from more than one scale [1]. In the case of our study, data from two scales are used, *i.e.*, transcriptome data and imaging data. For clarity, in the Introduction section we have added the following text:

"In multiscale modelling of biological systems, the term multiscale refers to the use of data from more than one scale [1]. We have used data from two scales, specifically transcriptome data and imaging data. Tumour subclone heterogeneity was identified at the genomic (transcriptome) scale using a fully unsupervised deconvolution method applied to gene expression profiles. Imaging scale heterogeneity was characterized using a set of radiomic feature analyses. The noninvasive radiogenomic signatures associating these two scales were further identified to classify patients into groups with distinct subclone compositions. Here, we investigated the biological and clinical relevance of modelling multiscale intratumour heterogeneity by conducting a radiogenomic analysis of 1,310 samples of breast cancer patients on five datasets from three data cohorts (Figures 1 and 2)."

1. Clarke, R., Tyson, J. J., Tan, M., Baumann, W. T., Xuan, J., & Wang, Y. (2019). Systems biology: perspectives on multiscale modeling in research on

endocrine-related cancers. *Endoc Relat Cancer*, 26, R345-R368.

2. I disagree with this sentence:

“Whether the molecular features of genomic subclones identified by unsupervised deconvolution of tumour gene expression can be used either to understand cancer biology and/or to predict prognosis accurately is largely unknown.” This has been extensively studied by several reports, in particular for deconvolution methods, these reports show clearly the correlation of deconvolved gene expression profiles with cancer biology, immune components and prognosis, also in breast cancer. The novelty in this study is really the use of a non-invasive biomarker that reflects the deconvolved components.

We greatly appreciate the insightful and supportive comment by the reviewer that *“The novelty in this study is really the use of a non-invasive biomarker that reflects the deconvolved components.”* To achieve this core objective, we have opted to apply an unsupervised deconvolution method, rather than the more widely used supervised approaches. There are three scientific considerations for taking this approach. First, we intend to show that unsupervised deconvolution can unbiasedly extract and confirm the presence of known predictive subclones (*e.g.*, our approach identified immune and cell cycle subclone components). Second, we wish to exploit additional yet novel/unknown predictive subclones, particularly when larger or better data sets become available in the foreseeable future; such subclones cannot be discovered using supervised methods. Third, while many known molecular signatures are available to support supervised deconvolution, these signatures are limiting and can be incomplete or even unreliable when the tumour microenvironment changes. For clarity, in the revised manuscript, we have modified the original sentence to now read:

“Whether the molecular signatures and/or compositions of genomic subclones (including novel ones) identified by unsupervised deconvolution of tumour gene expression profiles can be used either to understand cancer biology and/or to support noninvasive imaging-based prognosis is underexplored.”

3. The cohort discussion is confusing in several instances in the manuscript:

- *Figure 2 shows 4 cohorts, and the methods section mentions 3 cohorts. Can the authors make it consistent throughout the manuscript how many cohorts there are?*
- *The introduction mentions “two datasets containing both gene expression and survival data”, please specify exactly which two data sets. As far as I can tell there is only 1 such data set.*
- *The TCIA-TCGA breast cancer samples are a subset of the TCGA cohort, can the authors explicitly mention in the results that the TCIA-TCGA cohort samples were removed from the TCGA cohort? These samples cannot be used twice, in Phase 1 and phase2?*

We apologize for the confusion regarding the descriptions of the datasets/cohorts in the manuscript. We have revised Figure 2 (as shown below) to clarify the nature of and relationship among the datasets/cohorts used in this study.

There are five non-overlapped datasets from three cohorts that were purposely organized and used in the three stages of our analytical framework (described in the methods). **Cohort 1** contains matched genomic and follow-up data from TCGA and was randomly divided into two datasets, *i.e.* the genomic development dataset and the genomic testing dataset. **Cohort 2** contains matched gene expression and DCE-MRI data and is the radiogenomic dataset. **Cohort 3** contains matched DCE-MRI and follow-up data and includes two datasets: the prognostic assessment dataset and the prognostic testing dataset.

In the further revised manuscript, we first summarize the datasets as follows:

“Here, we investigated the biological and clinical relevance of multiscale intratumour heterogeneity modelling by conducting a radiogenomic analysis of 1,310 samples of breast cancer patients on five datasets from three data cohorts (Figures 1 and 2).”

Figure 2. Data organization flowchart

We then revised the description of the two datasets in Cohort 1 to read:

“...prognostic genomic signatures were identified using a genomic development dataset and tested on a genomic testing dataset. These two datasets are from Cohort 1, containing matched gene expression and survival data.”

In the revised Results section, we now state:

“Grouped samples were used independently (exclusively) in different stages/steps of this study (Fig. 2).”

The details about sample removal (as a data organization step) is discussed in Dataset section in Methods: “Samples with matched imaging and genomic data were included in Cohort 2 (n=87, after proper quality control procedure) and the remaining data (n=1010) were included in Cohort 1.”

4. I do not agree with the novelty in this statement: “Taken together, these results

show that an unsupervised gene expression deconvolution method can discover novel subclones associated with cancer-related pathways.” Most of these pathways have been reported at length for breast cancer, as the literature on gene signatures that are prognostic for breast cancer is very extensive and has already shown that the immune component is an important aspect for breast cancer survival prediction together with the obvious cell cycle pathways.

For clarity, we have revised this sentence to read:

“Taken together, these results show that an unsupervised deconvolution of gene expression data can unbiasedly extract and confirm several predictive subclones that are associated with known cancer-related pathways.”

5. Further, do the authors have any explanation, rationale, interpretation for the Alzheimer’s disease component, the peroxisome component, the ECM- component and the cholinergic synapse component? These are all components that have higher importance than the primary immunodeficiency component, yet are not discussed? Especially the Alzheimer’s component is puzzling. Any thoughts what the significance is of these components in breast cancer?

The reviewer has identified a common yet unresolved problem with current molecular pathway annotation and gene set enrichment analysis tools, an issue we have studied and discussed previously (in part) [1-3]. Briefly, the annotation of genes and pathways (gene set enrichment analysis) *e.g.*, as canonical signalling features or as being associated with a specific disease or cellular function, is generally achieved by a somewhat subjective analysis of often an extensive body of published literature. These tools rarely (if ever) account for the cellular context of the data and so are best used as generic guides. For example, a single input gene list may identify many different pathways where many of the same genes are included in (and therefore contribute to the selection of) more than one pathway, cellular function, or disease. Annotation of a signalling feature as being, *e.g.*, “Alzheimer’s”, generally reflects the contextual frequency with which this feature is reported in the literature. While the data in our study (as in all other studies that use these tools) provides the cellular context (breast cancer), we have little choice but to use the annotations used by the tool developers, even when this can appear a little misleading if interpreted literally.

1. Clarke, R., Tyson, J. J., Tan, M., Baumann, W. T., Xuan, J., & Wang, Y. (2019). Systems biology: perspectives on multiscale modeling in research on endocrine-related cancers. *Endoc Relat Cancer*, 26, R345-R368.

2. Clarke, R., Kraikivski, P., Jones, B. C., Sevigny, C. M., Sengupta, S., & Wang, Y. (2020). A systems biology approach to discovering pathway signaling dysregulation in metastasis. *Cancer Metastasis Reviews*, in press, 2020

3. Lu Y, Chang Y-T, Hoffman EP, Yu G, Herrington DM, Clarke R, Wu C-T, Chen L, Wang Y: Integrated Identification of Disease Specific Pathways Using Multi-omics data. *bioRxiv* 2019:666065.

Since we do not have the space, nor is it directly relevant to our study, to discuss this issue in any detail here, we have focused on how the individual subclone genes in any selected pathway may either affect, or correlate with, any relevant cancer-specific function(s). In the Discussion section we have added the following:

“The Alzheimer’s diseases subclone includes pathways with mitochondrial genes as annotated by the pathway analysis tools. Down-regulation of the NDUFA13 gene in the pathways has been shown to correlate with lymph node metastasis and advanced tumour-node-metastasis (TNM) stage in breast cancer [1]. The peroxisome subclone includes peroxisome, prostate cancer, pathways in cancer, metabolic pathways. Among these, the peroxisome pathway is reported as enriched among all breast tumour subtypes [2] and is correlated with cell proliferation and tumorigenesis [3, 4]. The extracellular matrix (ECM)-receptor interaction subclone includes ECM-receptor pathway, which is associated with progression and resistance to cytotoxic and hormonal treatments in breast cancer [5]. Moreover, the dysregulated PI3K-AKT signalling pathways in this subclone are often activated by mutations in breast cancers and so can be targeted by drugs [6]. The ECM-receptor interaction and the protein digestion and absorption pathways are reported as significantly enriched in analysis of differentially expressed genes between tumour tissues and paracancerous tissues [7]. The cholinergic synapse subclone includes key gene of BCL2, which is an established prognostic biomarker in early breast cancer [8].”

1. Zhou T, Chao L, Rong G, Wang C, Ma R, Wang X: Down-regulation of GRIM-19 is associated with STAT3 overexpression in breast carcinomas. *Hum Pathol* 2013, 44(9):1773-1779.
2. Wilson HE, Stanton DA, Montgomery C, Infante AM, Taylor M, Hazard-Jenkins H, Pugacheva EN, Pistilli EE: Skeletal muscle reprogramming by breast cancer regardless of treatment history or tumor molecular subtype. *NPJ breast cancer* 2020, 6:18.
3. Ham SA, Kim E, Yoo T, Lee WJ, Youn JH, Choi MJ, Han SG, Lee CH, Paek KS, Hwang JS *et al*: Ligand-activated interaction of PPARdelta with c-Myc governs the tumorigenicity of breast cancer. *Int J Cancer* 2018, 143(11):2985-2996.
4. Peters JM, Shah YM, Gonzalez FJ: The role of peroxisome proliferator-activated receptors in carcinogenesis and chemoprevention. *Nature reviews Cancer* 2012, 12(3):181-195.
5. Giussani M, Merlino G, Cappelletti V, Tagliabue E, Daidone MG: Tumor-extracellular matrix interactions: Identification of tools associated with breast cancer progression. *Seminars in cancer biology* 2015, 35:3-10.
6. Yang J, Nie J, Ma X, Wei Y, Peng Y, Wei X: Targeting PI3K in cancer: mechanisms and advances in clinical trials. *Mol Cancer* 2019, 18(1):26.
7. Bao Y, Wang L, Shi L, Yun F, Liu X, Chen Y, Chen C, Ren Y, Jia Y: Transcriptome profiling revealed multiple genes and ECM-receptor interaction pathways that may be associated with breast cancer. *Cell Mol Biol Lett* 2019, 24:38.

8. Dawson SJ, Makretsov N, Blows FM, Driver KE, Provenzano E, Le Quesne J, Baglietto L, Severi G, Giles GG, McLean C *et al*: BCL2 in breast cancer: a favourable prognostic marker across molecular subtypes and independent of adjuvant therapy received. *Brit J Cancer* 2010, 103(5):668-675.

6. I disagree that the TCGA data (this includes the TCGA cases that are used in cohort 2 that also have imaging data) has adequate survival events. For breast cancer it has been shown in several reports that 5 to 10 year follow-up data is required to be able to capture robust survival signatures across patients. The authors should mention the absolute number of censored patients and the average follow-up time for both censored and non-censored patients so the reader can judge for themselves. Based on the data that you report, 84% of patients in both the development and testing cohorts are censored, what are the follow-up times for these?

To allow readers to make their own determinations, we have followed the suggestion of the reviewer and provided the following additional information in the Dataset section in the Methods: “Within the genomic development dataset (n=660), 567 censored patients had a mean follow-up time of 3.147 years; the remaining patients (n=93) had a mean follow-up time of 3.769 years. For the genomic testing dataset (n=329), 283 patients were censored with a mean follow-up time of 3.150 years and the remaining patients (n=46) had mean follow-up time of 3.801 years. Cohort 2 (radiogenomic dataset, n=87) contains matched imaging and genomic data and was used to establish the associations between radiomic features and the prognostic genomic signatures (survival data are not needed).”

We also added the statement and illustration about the follow-up times provided in TCGA: “The follow-up time for data from TCGA (Cohorts 1 and 2) is shorter than that of Cohort 3 but we do not see a significant impact on the main outcomes of this work. In our study, Cohorts 1 and 2 are used solely to identify the prognostic radiogenomic features by associating the multiscale key components of intratumour heterogeneity, that contributes mechanistically to the poor outcomes in patients.” Please refer to Discussion section for details.

7. Figure 4 d: this panel is not clear, which one is it? One of the clones?

Figure 4d shows the Kaplan-Meier curves of overall survival (OS) between the ‘good survival’ and ‘poor survival’ groups in the genomic testing dataset.

In the genomic testing dataset, the gene expression profile was deconvoluted using the reference matrix obtained from the genomic development dataset, which generated the subclone fraction values. Patients were clustered into significantly different ‘good survival’ (n=153) and ‘poor survival’ (n=176) groups (Figure 4d, p=0.036) using the proportional compositions of key predictive subclones. As suggested, we have added clarity to the legend for Figure 4d, which now reads as:

“**d**) Kaplan-Meier curves of OS between ‘good survival’ (n=153) and ‘poor survival’ (n=176) groups in the genomic testing dataset.”

8. In the last paragraph of Phase 3, the p-value for RFS (p=0.037) does not match the p-value mentioned in Table 3

Thank you for pointing this out and we apologize for the mistake. The p-value for RFS should be $p=0.0037$ in the last paragraph of Phase 3, as shown in the Table 3. We have corrected this error.